# Structural basis for substrate gripping and translocation by the ClpB AAA+ disaggregase

Alexandrea N. Rizo[1,2], JiaBei Lin[3], Stephanie N. Gates[1], Eric Tse [ID][2], Stephen M. Bart[3], Laura M. Castellano[3], Frank DiMaio[4], James Shorter[3] & Daniel R. Southworth[2]

Bacterial ClpB and yeast Hsp104 are homologous Hsp100 protein disaggregases that serve critical functions in proteostasis by solubilizing protein aggregates. Two AAA+ nucleotide binding domains (NBDs) power polypeptide translocation through a central channel comprised of a hexameric spiral of protomers that contact substrate via conserved pore-loop interactions. Here we report cryo-EM structures of a hyperactive ClpB variant bound to the model substrate, casein in the presence of slowly hydrolysable ATPγS, which reveal the translocation mechanism. Distinct substrate-gripping interactions are identified for NBD1 and NBD2 pore loops. A trimer of N-terminal domains define a channel entrance that binds the polypeptide substrate adjacent to the topmost NBD1 contact. NBD conformations at the seam interface reveal how ATP hydrolysis-driven substrate disengagement and re-binding are precisely tuned to drive a directional, stepwise translocation cycle.

---

[1] Graduate Program in Chemical Biology, University of Michigan, Ann Arbor, MI 48109, USA. [2] Department of Biochemistry and Biophysics, Institute for Neurodegenerative Diseases, University of California, San Francisco, CA 94158, USA. [3] Department of Biochemistry and Biophysics, Perelman School of Medicine at the University of Pennsylvania, Philadelphia, PA 19104, USA. [4] Department of Biochemistry, University of Washington, Seattle, WA 98195, USA. Correspondence and requests for materials should be addressed to D.R.S. (email: daniel.southworth@ucsf.edu)

Heat-shock protein (Hsp) 100 protein complexes are critical cell stress responders that solubilize and unfold toxic protein aggregates and amyloids, thereby enhancing thermal and chemical tolerances[1,2]. They are members of the conserved family of AAA+ molecular machines that form dynamic hexameric-ring structures and undergo ATP hydrolysis-driven translocation of polypeptide substrates through a central channel[3–6]. Homologs ClpB from bacteria[7] and Hsp104 from yeast[8] are powered by two AAA+ nucleotide-binding domains (NBD1 and NBD2) per protomer and collaborate with the Hsp70 chaperone system to promote disaggregation and downstream refolding of substrates[9,10]. Hsp104 recognizes and unfolds amyloid structures and is required for the transmission of yeast prions, such as Sup35, thereby enhancing prion propagation and enabling selective advantages[11–13]. Pore-loop motifs within the AAA+ domains are essential for translocation and contain key Tyr residues that are a part of a highly conserved aromatic/hydrophobic pair proposed to interact and grip polypeptide substrates[14–17]. ATP hydrolysis requires conserved "Walker A" and "Walker B" motifs and an Arg-finger residue in an adjacent protomer that contacts the nucleotide and contributes to inter-domain communication[18].

An α-helical coiled-coil middle domain (MD) that wraps equatorially around the hexamer[8,19–22] interacts with Hsp70 during substrate loading[23,24] and modulates disaggregase function[25,26] likely through nucleotide-dependent conformational changes[6,20,27]. Single missense mutations in the MD can alter ATP hydrolysis and potentiate activity[27]. Indeed, MD variants have been identified that enable Hsp104 to more effectively dissolve self-templating fibrils formed by human neurodegenerative disease proteins including α-synuclein, TDP43, FUS, and TAF15[25,28–32]. In addition, amino-terminal domains (NTDs) are connected to the NBD1 by a flexible linker and form an additional ring[33] in the hexamer that is important for interaction with some substrates[7,19,34–36].

Recent cryo-EM structures of substrate-bound Hsp104[6] and ClpB[5,37] identify an asymmetric spiral architecture of the hexamer that stabilizes polypeptide substrate in the channel via pore-loop contacts from five protomers, while a sixth protomer at the interface of the spiral is unbound. This architecture is similar to other recent structures of single and double-ring AAA+ complexes[38–41], supporting a conserved substrate interaction mechanism. For Hsp104, we identified an additional substrate-bound conformation in which all six protomers contact substrate in a complete spiral, revealing a two amino acid-step translocation mechanism[6]. Additional structures of substrate-free hexameric states[6,20,22] identify an open "lock washer" conformation of the hexamer that may function during substrate engagement or release[6]. Previous structures have relied on Walker B mutations in both NBDs (DWB) that are stabilized for substrate and nucleotide-binding but inactive for hydrolysis and disaggregation[5,42,43], leading to questions about which conformations represent active states during translocation. Indeed, in a recent cryo-EM analysis of Hsp104[DWB] these conformations are proposed to support a stochastic mechanism[43]. However, considering the DWB likely uncouples conformational changes from the hydrolysis cycling that is required for disaggregation, how ATP hydrolysis synchronizes distinct hexamer conformations to drive translocation remains a key question.

To elucidate the disaggregation mechanism, we sought to determine cryo-EM structures of a substrate-bound ClpB complex from *Escherichia coli* that is active for ATP hydrolysis and polypeptide translocation. The ClpB[K476C] MD variant was chosen due to its established hyperactive function[27]. Utilizing ATPγS, we determined the structure of ClpB[K476C] bound to the substrate casein to 2.9 Å resolution. An array of substrate contacts are precisely defined, revealing distinct substrate interaction mechanisms coordinated by different NBD1 and NBD2 pore-loop motifs along the channel. Modeling of the well-resolved substrate density reveals specific sequence characteristics that are stabilized by NBD1 and NBD2. Refinement of the NTD ring revels a trimer of alternating N-terminal domains that form a substrate entrance channel and position the polypeptide above NBD1. Finally, we identify NBD1 and NBD2 conformational changes at the seam interface that coincide with changes in nucleotide state and substrate release, indicating how coordinated hydrolysis across the NBDs drives a directional translocation cycle.

## Results

**Substrate-bound structure of a ClpB hyperactive variant.** Similar to previous studies[5,6], fluorescein-labeled (FITC) casein and ATPγS, a slowly hydrolyzable analog that can power translocation of unfolded polypeptides in vitro[44,45], were used to investigate active, substrate-bound complexes. WT ClpB was initially tested and forms a substrate-bound complex; however, reconstructions went to a modest, 5.7 Å resolution, indicating hexamer stability or heterogeneity may be present (Supplementary Fig. 1a, b; Supplementary Table 1). To identify a stable, but active complex, the hyperactive ClpB variant containing a K476C mutation in helix L2 of the MD[27] was tested. ClpB[K476C] bound substrate similarly to WT and 2D reference-free averages of the fractionated WT- and K476C-substrate-bound samples show similar hexamer conformations compared to previous structures[6] (Supplementary Fig. 1a–c).

Compared to WT, ClpB[K476C] displayed ~2-fold elevated ATPase activity in the absence of casein and ~5-fold elevated ATPase activity in the presence of casein (Fig. 1a). Moreover, in the presence of ATP but without the Hsp70 chaperone system (DnaK, DnaJ, and GrpE [KJE]), ClpB[K476C] displayed substantially elevated luciferase disaggregase activity compared to WT (Fig. 1a). WT and ClpB[K476C] displayed similar enhanced disaggregase activity in the presence of KJE. Luciferase disaggregation was not observed above background when ATP was replaced with ATPγS for WT and ClpB[K476C]. However, ClpB has been shown to translocate soluble, unfolded polypeptides like casein in the presence of ATPγS[45]. WT and ClpB[K476C] were determined to bind casein equivalently and with high affinity (apparent $K_d = 130$ and 110 nM, respectively) in the presence of ATPγS (Supplementary Fig. 1d).

The ClpB[K476C] casein-bound complex, incubated with ATPγS and purified by SEC, was targeted for high-resolution structure determination by cryo-EM. The final map refined to an indicated 2.9 Å overall resolution using the total dataset following sorting by 2D classification (Supplementary Fig. 1e; Supplementary Table 1). The channel and substrate-bound protomers are at the highest resolution for the complex (<3.0 Å), while the seam protomers are lower resolution (~4.0 Å), indicating flexibility at this interface (Fig. 1b). The angular distribution of the particles show top and side-view preferred orientations, and the 2D projections of the map exhibit well-defined features that match the averages (Supplementary Fig. 1f, g). The protomers (P1–P6) adopt a right-handed spiral configuration similar to previously described structures[5,6], and polypeptide density is identified to span the 80 Å length of NBD1 and NBD2 that comprise the translocation channel (Fig. 1c). Density for the AAA+ domains show well-defined features indicative of the resolution (Supplementary Fig. 1h, i). Density along the channel, attributed to the bound substrate, was modeled initially with a 26-residue, unfolded strand of poly-Ala (Fig. 1d). The ClpB[K476C] map and model were compared with our lower-resolution ClpB[WT] map and determined to have a similar architecture (Supplementary

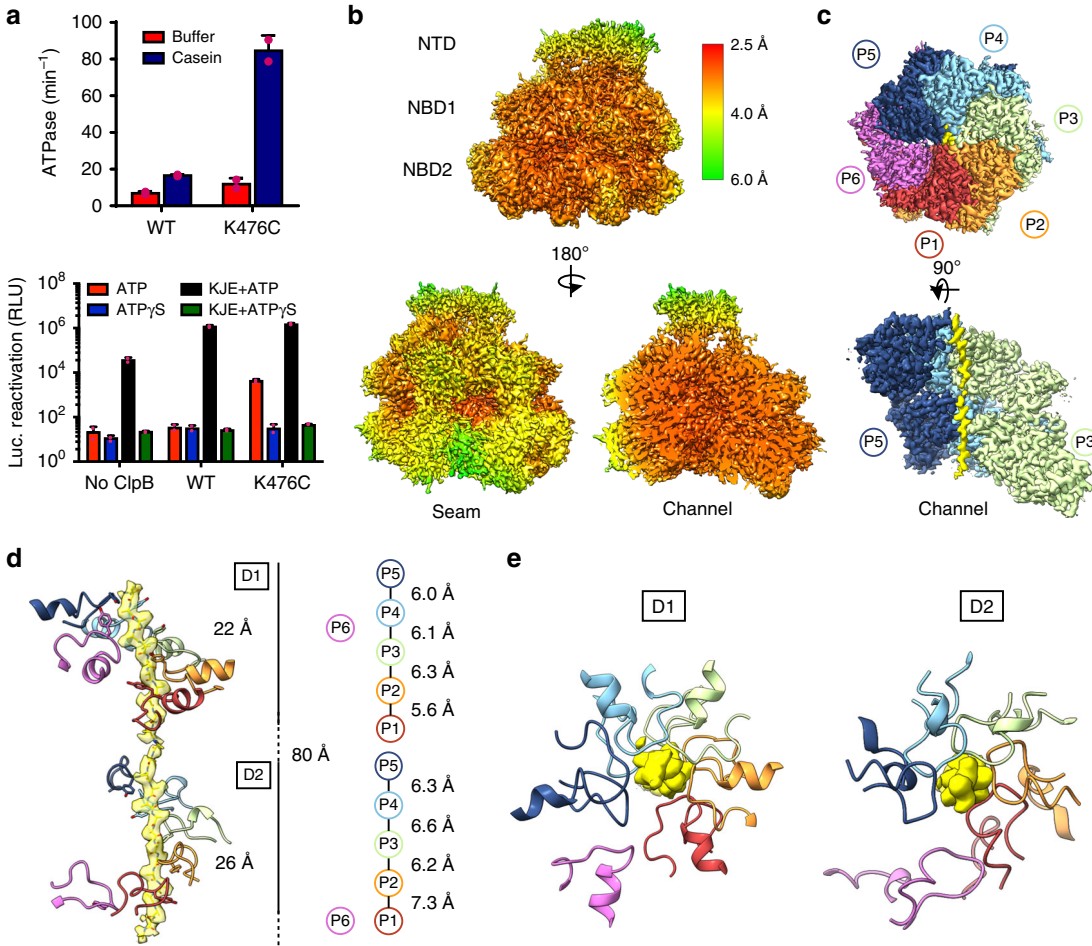

**Fig. 1** Activity and substrate-bound structure of the ClpB[K476C] hyperactive variant. **a** ClpB[WT] and ClpB[K476C] ATPase activity (upper panel) measured in the presence or absence of casein. Y-axis values are of phosphate release (min[−1]) and represent means ± SD. ClpB[WT] and ClpB[K476C] luciferase disaggregase activity (lower panel), plotted as relative luminescence, measured in the absence or presence of KJE with either ATP or ATPγS. Values represent means ± SD. For both plots, $n = 2$ and the data points are shown in magenta. EM density map of the ATPγS-ClpB[K476C]:casein complex **b** colored by resolution[74] and **c** colored to show individual protomers (P1–P6) and substrate (yellow). **d** Side and **e** top-views of the NBD1 and NBD2 Tyr-containing pore loops (colored by protomer) with substrate EM density (yellow) modeled with a 26-residue poly-Ala. A schematic is shown with the distances (Å) between Tyr-substrate contacts along the NBD1 and NBD2 for protomers P1–P5; protomer P6 (magenta) is disconnected from the substrate

Fig. 1j, k). Notably, while density for the MD is not well-defined in the final maps, lower-resolution density corresponding to the MD coiled-coil is observed for protomers P3–P5 in certain classes and modeling the MD motif 2 into the density confirms that in these classes the MD adopts an ATP-state conformation for these protomers (Supplementary Fig. 1l). This arrangement matches the MD conformation identified in Hsp104[6,20] and *Mycobacterium tuberculosis* ClpB[37].

In the ClpB[K476C]:casein structure five protomers directly contact the substrate, which is positioned slightly off center to the channel axis and closer to the back protomers (P2–P5), opposite to the seam interface (Fig. 1d, e). The conserved NBD1 and NBD2 Tyr-containing pore loops[3,16] are each separated by ~6–7 Å along the substrate and rotate ~60° around this axis (Fig. 1d). This arrangement indicates an overall dipeptide spacing similar to previous substrate-bound AAA+ structures[5,6], with the Tyr residues (Y251 for NBD1 and Y653 for NBD2) intercalating between substrate side chains, directly contacting the backbone. The more flexible seam protomer, P6, which is adjacent to the highest (P5) and lowest (P1) contact positions is asymmetric and disconnected from the substrate, with the NBD1 and NBD2 pore loops positioned ~4–5 Å away (Fig. 1d, e).

**Distinct substrate gripping interactions by NBD1 and NBD2.** The resolution of the substrate channel is improved compared to previous structures of ClpB and Hsp104, and enables precise mapping of the pore-loop interactions. Two distinct NBD1 loops (D1, residues 247–258 and D1′, residues 284–295) extend into the channel and contact the substrate (Fig. 2a; Supplementary Fig. 2a). In the canonical pore loop (D1), Y251 is flanked by basic residues in ClpB and Hsp104 (K250 and R252 in ClpB and K256 and K258 in Hsp104) (Fig. 2b). This pattern is distinct from the well-characterized aromatic-hydrophobic pair motif (Ar/Φ) (e.g. Tyr–Val) found in other AAA+ translocases and domains, including NBD2 of ClpB and Hsp104[46]. In this structure, we identify that K250 and R252 extend perpendicular to the substrate axis and together with Y251 form a well-defined clamp around the polypeptide (Fig. 2c). The side chains of K250 and R252 extend to contact E254 and E256 in the neighboring lower and upper pore loops, respectively (Fig. 2d). Based on the orientation and distance between the side chains, these residues likely support salt bridge or hydrogen bonding across the pore loops in all protomers except at the seam interface (with protomer P6), thereby stabilizing the flexible pore loops to support the dipeptide spacing.

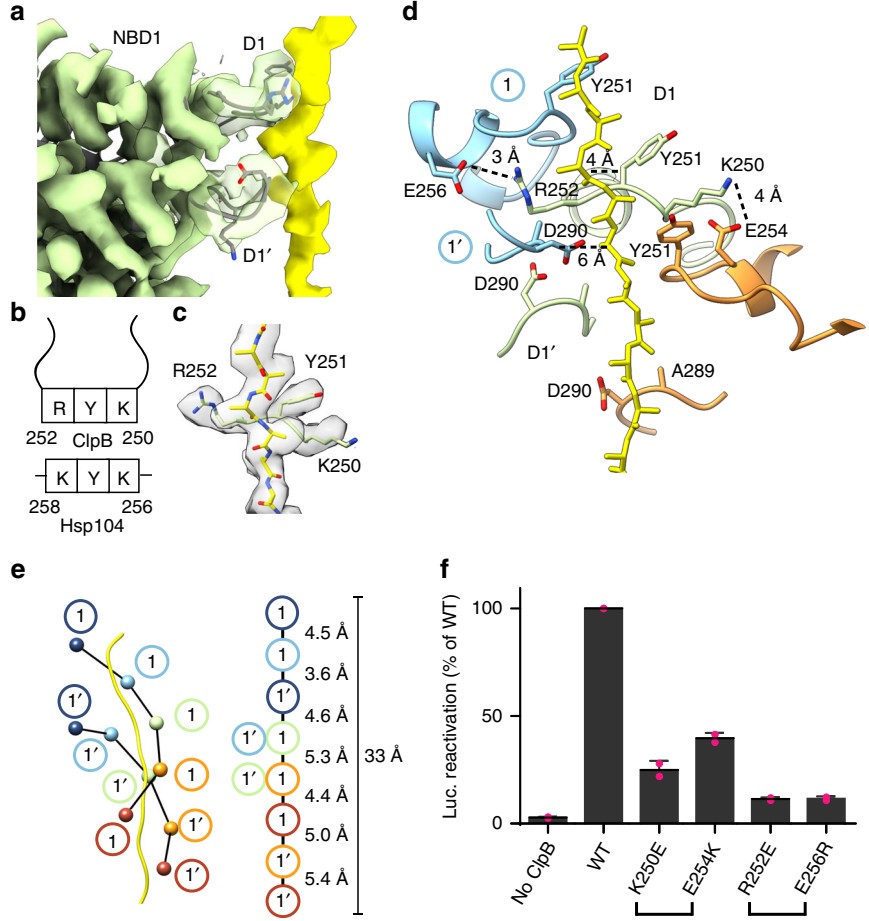

**Fig. 2** Substrate interactions by the NBD1. **a** Side view of the P3 NBD1 pore loop–substrate interactions with the D1 and D1′ loops indicated. **b** Schematic representation of the conserved D1 loop residues in ClpB and Hsp104. **c** Map and model of P3 D1 pore loop showing arrangement of K250, Y251, and R252 along the substrate density modeled with poly-Ala (yellow). **d** P2–P4 D1 and D1′ pore loops and modeled poly-Ala substrate with substrate interactions and proposed salt bridge interactions: E256-R252 and K250-E254 shown with approximate distances (dotted line). **e** Schematic of the double spiral of substrate interactions that is formed from the D1 and D1′ loops, with distances along the substrate axis shown based on the position of Y251 in the D1 and D290 in the D1′ pore loops (right). **f** Luciferase disaggregase activity was measured for WT, ClpB$^{K250E}$, ClpB$^{R252E}$, ClpB$^{E254K}$, or ClpB$^{E256R}$ in the presence of KJE plus ATP. Values represent means ± SD ($n = 2$, data points shown in magenta)

The secondary pore loops (D1′) are more variable around the hexamer; however, A289 and D290 are consistently oriented toward the substrate and ~6 Å away from the backbone (Fig. 2d). Thus, these loops form an additional set of substrate contacts that are shifted ~8 Å down the axis from the D1 loops (Fig. 2e). Together the D1 and D1′ loops enable an ~11 amino acid-length (33 Å) of the polypeptide substrate to be stabilized by NBD1. While the D1′ loops increase the overall interactions with substrate made by NBD1, given the variability and distance from the substrate, their contributions are likely smaller compared to binding by the KYR motif in the D1 loops.

The functional significance of the cross pore loop interactions was assessed by charge reversal mutations K250E, R252E, E254K, and E256R. All point mutations resulted in substantially reduced affinity for binding to casein (Supplementary Fig. 2b), and were defective in luciferase reactivation in the presence of KJE (Fig. 2f), indicating that these contacts are critical for substrate binding and translocation. These NBD1 residues are highly conserved among ClpB homologs (Supplementary Fig. 2c) and may enable distinct substrate-binding functions compared to the NBD2.

The NBD2 also contains two pore-loop strands in each protomer that extend and contact the substrate polypeptide for P1–P5 (Fig. 3a; Supplementary Fig. 3). These are the canonical loop (D2, residues 647–660) containing the conserved Ar/Φ

(Y653 and V654) motif and an additional short helical loop (D2′, residues 636–646). In the D2 loop, Y653 and V654 together form a clamp around the substrate backbone, intercalating between the side chains, with both residues appearing to make similar contributions to substrate binding (Fig. 3b). For the D2′ loop residues E639, K640, and H641 project into the channel and appear to interact directly with the substrate polypeptide (Fig. 3c). E639 and H641, in particular, are adjacent to the side chains of the substrate at certain positions, with H641 positioned between the Y653 residues of two clockwise protomers. Thus, the D2′ loops form an additional spiral of substrate interactions that coalesce toward the bottom of the NBD2 where the polypeptide exits from the translocation channel (Fig. 3d). To assess function of the D2′ loop, point mutations E639K, K640A, K640E, H641A, and H461E were tested for disaggregase activity. All mutations showed loss of activity relative to WT (Fig. 3e) with ClpB$^{E639K}$ and ClpB$^{K640E}$ at ~40% of WT, whereas mutations at H641 (to A or E) are ~60% of WT (Fig. 3e). Thus, substrate contact by these D2′ loop residues is likely important for substrate translocation.

**Modeling substrate sequences across NBD1 and NBD2.** The cryo-EM density corresponding to the casein substrate polypeptide is well-defined across the NBD1 and NBD2 portions of

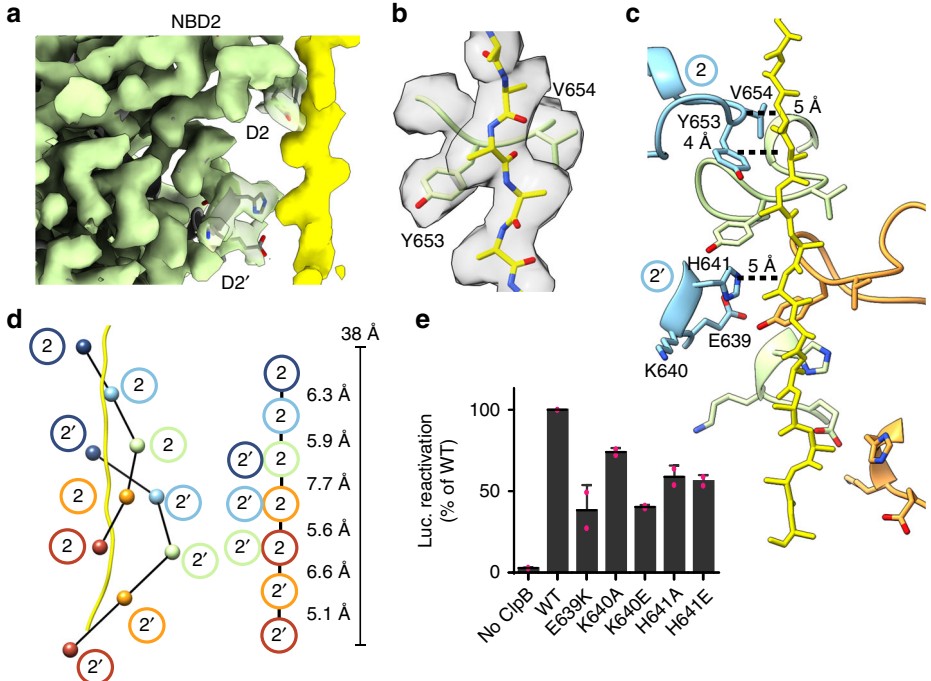

**Fig. 3** Substrate interactions by the NBD2. **a** Side view of the P3 NBD2 pore loop-substrate interactions with the D2 and D2′ loops indicated. **b** Map and model of P3 D2 pore loop showing arrangement of Y653 and V654, along the substrate density modeled with poly-Ala (yellow). **c** The D2 and D2′ pore loops of P2–P4, colored by protomer, and substrate (yellow), with distances between interacting residues and substrate backbone shown. **d** Schematic of D2 and D2′ loops interacting along the substrate for protomers P1–P5 with distances along the substrate axis shown based on the position of Y653 and H641 for the D2 and D2′, respectively. **e** ClpB, ClpB[E639K], ClpB[K640A], ClpB[K640E], ClpB[H641A], or ClpB[H641E] luciferase disaggregase activity was measured in the presence of KJE plus ATP. Values represent means ± SD ($n = 2$)

the channel and the regularly spaced densities extending from the backbone are consistent with side chains based on the fitting of a poly-Ala model (Fig. 1d). Thus, ClpB may interact preferentially with certain sequences of casein, leading to improved resolution of the substrate. Additionally, sequence-specific interactions may be enhanced by the presence of ATPγS, given the affinity for casein is substantially higher compared to when ATP is present[6].

To identify optimal sequences that have an improved fit to the polypeptide density compared to poly-Ala, 1604 peptides of casein, corresponding to overlapping sequences from the four isoforms, α-s1, α-s2, β, and κ (Supplementary Table 2), were threaded into the density for the NBD1 and NBD2 domains using Rosetta[47]. The model for the peptides with the lowest energy scores relative to poly-Ala are shown (Fig. 4a, b) and comprise casein sequences: VVTILALTLPF (κ isoform, residues 8–18) for the NBD1 and ILACLVALALA (β isoform, residues 5–15) for the NBD2. Similar low-energy scores were achieved for additional peptides that span the casein sequences tested, while many peptides scored unfavorably compared to poly-Ala (Supplementary Fig. 4; Supplementary Table 3). Amino acid preferences at each position were ranked relative to Ala (Fig. 4c, d). This ranking was achieved by taking the optimal casein peptides (above) and individually mutating each of the residues to all 20 amino acids, then comparing the energies of the fit at these mutated positions in arrangements consistent with the peptide backbone density. Energetically favorable interactions are identified primarily for large aromatic and hydrophobic residues for both NBD1 and NBD2, while residues with small sidechains, such as Pro and Gly, interact unfavorably (Fig. 4c, d). Particularly for NBD1, the lower energy, favorable interactions alternate along the peptide sequence with the substrate side-chain residues that stack between the conserved Tyr residues in the model (Fig. 4a, b).

Notably, interactions by NBD1 are lower energy and more specific compared to the NBD2, likely reflecting the differences in the pore loop interactions discussed above (KYR for the NBD1 compared to YV for the NBD2) (Fig. 4c, d). These results agree with previous studies identifying that ClpB binds to peptides in enriched in aromatic residues[16]. Moreover, these findings reveal how ClpB can readily accommodate residues with bulky side chains and may interact preferentially with certain hydrophobic substrate sequences during disaggregation.

### A trimer of NTDs engages substrate.

The polypeptide substrate is expected to transfer to the AAA+ NBD1–NBD2 translocation channel via the NTDs[3], which form an additional ring adjacent the NBD1 ring[5,6] (Fig. 1b). However, this transfer has never been observed directly and the resolution of the NTDs is typically lower due to its flexibility. To characterize the NTD ring in the ClpB[K476C]:casein structure, particle re-centering and focused classification and refinement was performed (Supplementary Fig. 5a). This improved the density for the individual NTDs and connecting NBD1 linker (Supplementary Fig. 5c, d) enabling the identification of a trimer NTDs form the channel entrance (Fig. 5a). NTDs from protomers P1, P3, and P5 were determined to form the NTD ring, which retains the overall spiral architecture of the NBDs (Fig. 5a). Additional NTD density corresponding to the other protomers (P2, P4, and P6) is not observed, indicating they are likely disconnected and flexible. Remarkably, density corresponding to the polypeptide substrate is identified to extend up from the P5-NBD1 pore loop and directly contact the P5 and P3 NTDs (Fig. 5a). An additional nine residues were modeled into this substrate density as poly-Ala. Thus, we identify approximately 35 substrate residues are stabilized in an unfolded arrangement along the channel (Fig. 5b). Based on

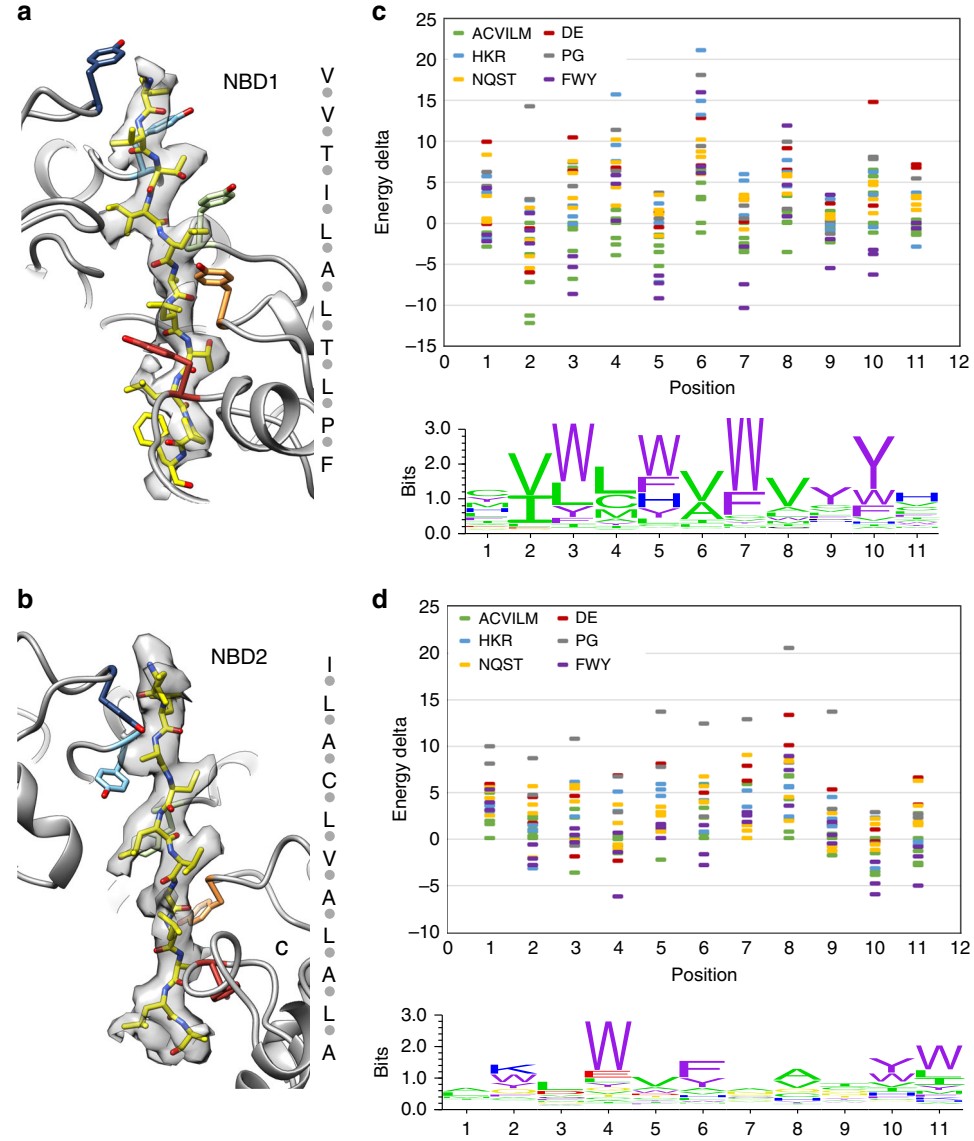

**Fig. 4** Analysis of the substrate sequence stabilized by the NBD1 and NBD2. Molecular model and sequence of the energetically favored casein peptide sequence (yellow) in density for the NBD1 (**a**) and NBD2 (**b**). Interacting Tyr resides (Y251 and Y653) are colored by protomer. A plot showing per-energy deviations at each of the 11 positions interacting with NBD1 (**c**) and NBD2 (**d**); the logo plots convert these energies to a distribution, showing the sequence preferences at each position

the molecular model, NTD-substrate interactions are proposed to involve helix A1, which contacts the substrate approximately two residues above P5-NBD1 pore loop in protomer P3 and ~7 residues above in protomer P5 (Fig. 5c). Interactions are also observed between helix A6 in protomer P3 and the substrate (Fig. 5b, c). In support of these interactions, residues in helices A1 and A6 that appear to be adjacent the substrate polypeptide in the structure were previously identified to form a substrate binding-groove and interact with hydrophobic regions of substrates[36] (Fig. 5c). However, additional conserved residues (T7, D103, E109) that have previously been proposed to interact with the substrate[48,49] appear more distal in our model. Thus, additional NTD conformations may be important during translocation. Overall, we identify that the NTD ring comprises three domains from alternating protomers, P1, P3, and P5, which together form an additional right-handed spiral and directly contact the substrate, establishing a direct role for the NTDs in substrate transfer to the AAA+ domains.

**Nucleotide-specific conformations reveal translocation steps.** Similar to previous substrate-bound structures[5,6,37], the protomers at the seam interface are identified to be flexible and at a lower overall resolution following refinement of the total dataset (after 2D classification) (Fig. 1b). Following 3D classification two distinct structures (Pre and Post states) are identified and refined to 3.4, and 3.7 Å, respectively (Supplementary Fig. 6a, c). Focus classification was performed to further improve the map of the P6–P1 interface (Fig. 6a; Supplementary Fig. 6b). The Pre-state P6–P1 protomer arrangement is identical to the model determined from refinement of the total dataset (discussed above) and some conformational differences are identified in comparison to Hsp104[6] and ClpB-BAP[DWB5] (Supplementary Fig. 6d).

The arrangement of the P6-P1 seam in the Post state involves substantial conformational changes of the NBDs in both protomers (Fig. 6b). In comparing Pre to Post state conformational changes, the NBD1 of P1 and P6 rotate together clockwise and upwards along the substrate axis (Fig. 6b, Supplementary

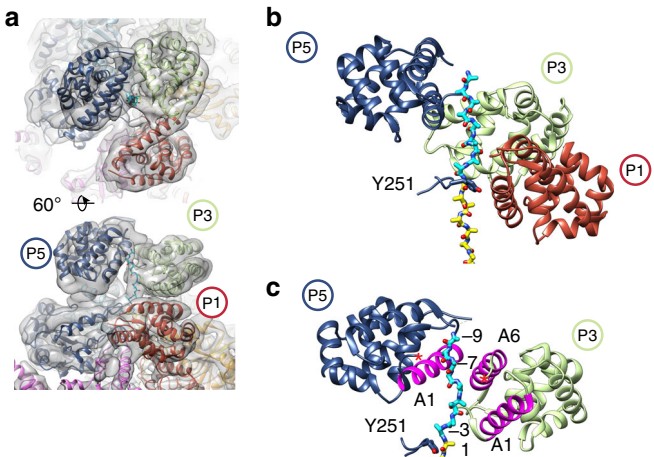

**Fig. 5** 3D focus classification reveals substrate interaction with NTDs. **a** Views of the NTD-focused refinement map after particle re-centering, colored by protomer, showing the P1–P3–P5 trimeric NTD ring. **b** Model of the P1–P3–P5 NTD trimer, and substrate polypeptide, modeled as poly-Ala (cyan), that extends from the P5 NBD1 pore loop (Y251). **c** Model of P5 and P3 NBDs interacting with substrate. Substrate-interacting helices A1 and A6 (magenta), previously characterized[36], are shown

Movie 1). The NBD2s similarly rotate upward and toward the substrate axis in a clockwise manner. The P1-NBD1 loop in the Post state shifts upward along the substrate axis by ~9 Å and becomes disconnected from the substrate and asymmetric to the helical position of the P2–P5 pore loops (Fig. 6c). The P6-NBD1 loop similarly shifts upward by ~7 Å, becoming nearly parallel with P5 at the top contact site, but remains disconnected from the substrate. Based on these comparisons, the Post state appears to be in an intermediate translocation state with respect to the NBD1 pore-loop positions, with protomer P6 on-path to bind next site along polypeptide, above the P5 position, while protomer P1 appears to be transitioning to a disconnected state, beginning with release by the NBD1 pore loop.

The nucleotide states of the NBD1 and NBD2 were assessed based on the density for ATPγS and the position of the Arg finger of the clockwise protomer, which contacts the γ-phosphate in the ATP-bound active state[18] (Supplementary Fig. 6e, f). Similar to previous structures[6], the NBD1 and NBD2 of the substrate-bound protomers, P3–P5, are bound to ATPγS and in an active configuration in both the Pre and Post states (Fig. 6d, Supplementary Fig. 6e). Conversely, the NBD1 and NBD2 of the seam protomers are in different nucleotide states and undergo changes between Pre and Post states that indicate hydrolysis is occurring (Supplementary Fig. 6f). Notably, the P1-NBD1 is bound to ATPγS and in an active state in the Pre-state conformation, but bound to ADP and inactive in the Post state (Supplementary Fig. 6f). This coincides with loss of substrate contact by the P1-NBD1 pore loop between the Pre and Post states (Fig. 6c). Conversely, the P1-NBD2 appears to be bound to ADP and inactive in both the Pre and Post states (Supplementary Fig. 6f). Additionally, the P6 NBDs, which do not contact substrate, are both identified to be inactive in the Pre and Post states. Notably, the NBD2 nucleotide pocket of P6 shifts from ADP-bound to apo based on the lack of density for nucleotide in the Post state. Finally, the NBD2 for protomer P2 likely shifts from ATP- to ADP-bound between the Pre and Post states (Supplementary Fig. 6f). Considering P2-NBD2 makes the next-to-last contact with the substrate, hydrolysis may be triggered initially at this site for a counterclockwise rotary cycle (Fig. 6c). Thus, while the stable, substrate-bound protomers (P3–P5)

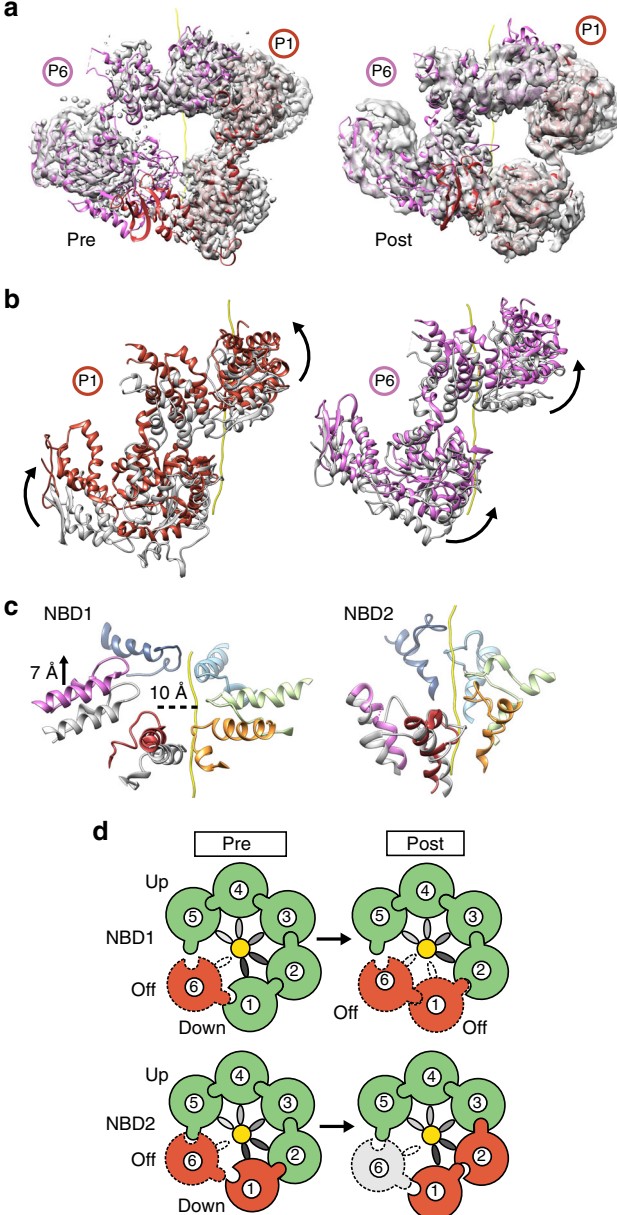

**Fig. 6** Conformational changes and nucleotide states of the protomers at the seam interface. **a** Post-process masked cryo-EM map and molecular model of protomers P1 and P6 in the "Pre" and "Post" conformational states following particle re-centering and focused refinement of the seam interface. **b** Protomers P1 and P6 for both Pre (gray) and Post (colored) states with substrate (yellow), shown superimposed following alignment to protomer P3 in the hexamer. Conformational changes are shown as distinct rotations (arrows) of the NBD1 and NBD2. **c** The Pre- (gray) and Post-state (colored) NBD1 and NBD2 pore loops shown superimposed for protomers P1 and P6 with distances indicated following alignment of P3 pore loops in the hexamer for both states. **d** Schematic showing proposed NBD1 and NBD2 nucleotide states of the hexamer for the Pre and Post states, colored to indicate ATP (green), ADP (red), and APO (gray) state, based on analysis of the nucleotide pockets in the two structures (Supplementary Fig. 6e, f). Protomers that are disconnected from the substrate are indicated with a dash and Arg-finger contact is indicated by the interlocking contact between two protomers

remain in an ATP-active state in the Pre and Post states, the protomers at the lower substrate contact sites and at the seam interface (P1, P2, and P6) undergo conformational and nucleotide state changes that are consistent with a hydrolysis-driven

substrate release mechanism that is sequential across NBD1 and NBD2 (Fig. 6d).

**NBD1–NBD2 rearrangement with hydrolysis and substrate release.** In side views of the reference-free 2D projection averages the NBD1 and NBD2 rings adopt a non-parallel configuration identified by an increase in the distance between the rings on one side versus the other (Fig. 7a). This asymmetry is identified in both Pre and Post-state hexamer models, in which the overall length of the NBD1–NBD2 double ring increases at the P6–P1 seam interface compared to the P3 and P4 protomers across the hexamer, going from ~75 to 106 Å in the Pre-state and ~77 to 96 Å in the Post-state (Fig. 7b). When the protomers are separated and individually aligned to the NBD1 large domain to normalize the axial rise, the separation between the NBD1 and

NBD2 is identified to be identical for the substrate-bound protomers (P2–P5), but increases upon moving to protomers P1 and P6 at the seam where substrate is released (Fig. 7c). Notably, the distance between residues I546 and P594 at the NBD1–NBD2 interface increases from 13 to ~30 Å across these protomer conformations. This distance also increases between the Pre and Post states for protomer P1, which undergoes substrate-release and conversion to an ADP state for NBD1 (compare Fig. 7c and Fig. 6). Alignment of the protomers reveals substantial NBD1–NBD2 conformational changes occur between these protomers, which primarily involve rotations of the NBD1–NBD2 connecting residues (545–555) (Supplementary Movie 2) and appear to coincide with substrate release and ATP hydrolysis (Fig. 7d). From these data we propose a model in which the NBD1–NBD2 connecting residues function as a nucleotide-driven swinging arm or spring that alters the positions of NBD1

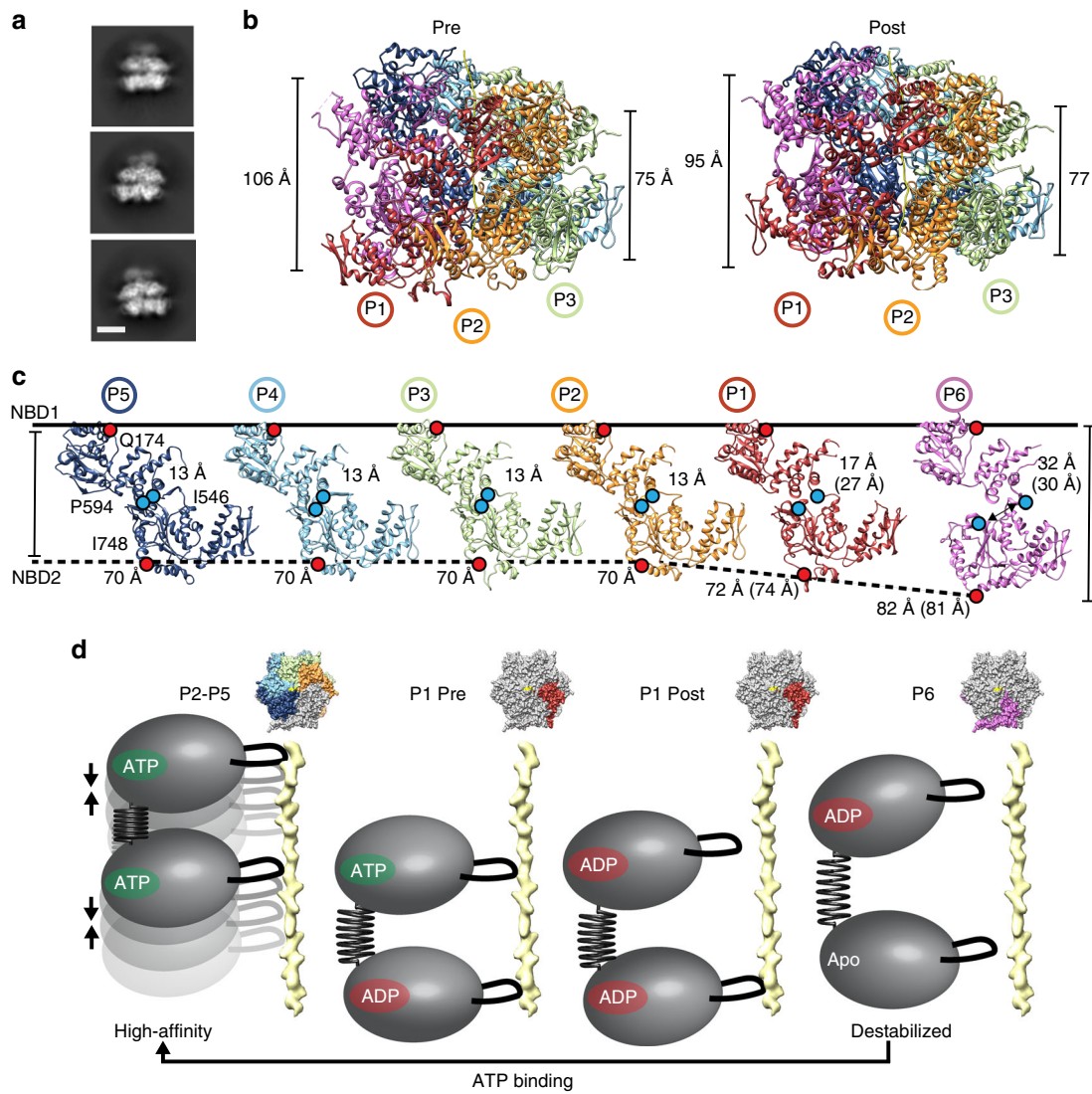

**Fig. 7** Expansion between NBD1 and NBD2 at the seam interface. **a** 2D reference-free class averages of hexamer side views showing non-parallel arrangement of the NBD1 and NBD2 AAA+ rings. The scale bar equals 50 Å. **b** Side view of the Pre- and Post-state models showing the expansion of NBD1–NBD2 rings measured at the P1–P6 seam interface (left) compared to the P3–P4 interface across the hexamer (right). **c** The individual protomers from the Pre-state shown separated and aligned to the NBD1. Distance measurements are shown for residues Q174 and I748 (red dots) to depict overall protomer changes, and for residues I546 and P594 (blue dots) to show changes at the NBD1–NBD2 interface. Distances for the P1 and P6 protomers in the Post state are shown in parentheses. **d** Model depicting ATP hydrolysis at the seam interface coincides with conformational changes and an expanded NBD1–NBD2 arrangement. ATP re-binding promotes a more compact NBD1–NBD2 conformation that favors high-affinity substrate interactions and interprotomer contacts with the clockwise, ATP-state protomer, thereby advancing substrate contacts to the next highest position and maintaining the unfolded state of the polypeptide. Model based on NBD1–NBD2 measurements in **c** and the nucleotide states in Supplementary Fig. 6e, f

and NBD2 during translocation to coordinate substrate-binding and release steps. When NBD1 and NBD2 are bound to ATP, the protomer conformation favors high-affinity substrate and inter-protomer interactions that function together across 4–5 proto-mers in the spiral to stabilize the polypeptide in an unfolded state. ATP hydrolysis, likely initiating in the NBD2 at the lowest sub-strate contact site, drives NBD1–NBD2 conformational changes which destabilize these interactions and trigger substrate release, resulting in the unbound conformation identified for protomer P6. Moreover, these conformational changes could be critical in establishing directionality in translocation wherein following ATP binding, the NBD1–NBD2 re-arranges to favor interaction with the clockwise, ATP-bound protomer at the topmost substrate position. This rearrangement would ensure translocation in the N-to-C direction down the channel, as opposed to re-binding to the protomer at the lowest position along the substrate (P1).

## Discussion

A number of recent structures of substrate-bound AAA+ trans-locases have begun to reveal a conserved mechanism of substrate interaction and translocation. For the Hsp100 disaggregases, key questions have remained about the specific roles of the two AAA+ domains, NBD1 and NBD2, and how conformational changes and ATP hydrolysis might be tuned in the different homologs to drive polypeptide translocation. To better elucidate the Hsp100 disaggregation mechanism we determined cryo-EM structures of the ClpB hyperactive variant, K476C, bound to the model sub-strate, casein. We focused on this variant because of its robust ATPase and disaggregase activity in order to capture active states of translocation. In the ATPγS-ClpB[K476C]:casein structure we identify a well-resolved translocation channel with a defined NBD1–NBD2 spiral of pore-loop-substrate contacts, revealing distinct substrate gripping mechanisms and sequence interaction specificity for the AAA+ domains. We identify a substrate channel entrance that comprises an NTD trimer which binds substrate as a spiral, extending its unfolded state, thereby facil-itating transfer to the AAA+ channel. Two conformations of the seam protomers are identified in different nucleotide states, revealing the translocation-step mechanism. Notably, the ~3.9 Å structures of ClpB from *M. tuberculosis*[37], published during review of this manuscript, are consistent with the hexamer architecture identified here. Notably, the NTD-trimer arrange-ment with the bound polypeptide substrate was not resolved in previous studies. Finally, we identify an expansion between the NBD1 and NBD2 occurs toward the seam interface and coincides with ATP hydrolysis and substrate release, revealing how ATP hydrolysis-driven conformational changes at this site may con-tribute to translocation cycling (Fig. 7d).

The NBDs of ClpB, Hsp104, and related Hsp100s are members of distinct clades of the AAA+ family. The NBD1 is a member of clade 3 which includes FtsH, p97, NSF, and katanin, while NBD2 is a member of clade 5 which includes ClpX, RuvB, and Lon[46,50]. Although evolutionarily distinct, the pore loops across these clades primarily contain the conserved Ar/Φ substrate-interacting motif, which is often flanked by Gly residues (e.g. Gly-Tyr-Val-Gly in *E. coli* ClpX, and the NBD2 of ClpB), indicating conserved functions[14]. The KYR motif, in which the substrate-interacting Tyr is flanked by basic residues, appears to be a unique feature of the NBD1 in disaggregases such as ClpB, Hsp104, and ClpA. Here we identify that K250 and R252 serve key roles in stabilizing the pore loops via salt bridge interactions with E254 and E256 in the pore loops of adjacent protomers (Fig. 2d). Importantly, we identify that charge reversal mutations at K250, R252, E254, or E256 results in a loss of activity in vitro, indicating these inter-actions are required for disaggregase function. Together with

Y251, these interactions likely contribute greater substrate-binding energy compared to the Ar/Φ motif of NBD2. Notably, Y251A has a less severe phenotype than Y653A in vivo[3]. Based on the structure here, the K250 and R252 cross pore-loop interac-tions as well as the NTD may partially compensate for this loss of function in the NBD1 compared to the NBD2. Considering the NBD1 makes the first contact with substrate beyond the more flexible NTD, these strong interactions may be critical for initi-ating substrate unfolding or facilitating processivity.

In NBD2, cross pore loop contacts are not present; however, Y653 and V654 form a well-defined clamp arrangement around the substrate backbone (Fig. 3b). Based on our modeling experiments, the substrate sequence preferences for NBD2 also favor aromatic and hydrophobic residues for positions that match the pore loop interactions. However, the energy change is not as substantial (Fig. 4c, d), indicating the NBD2 may exhibit weaker, and more nonspecific interactions compared to the NBD1. Thus, the NBD1 and NBD2 have distinct mechanisms for substrate gripping that likely reflect specific roles in disaggregation. Nota-bly, we also identify contacts by secondary pore loops (D1′ and D2′), supporting roles for these additional residues in the channel in further stabilizing the substrate in an unfolded arrangement. In particular, we identify residues H641 and E639 in the lower portion of the NBD2 channel comprise the majority of contacts where substrate would exit the translocation channel.

By modeling peptides into the substrate density we identify sequences enriched in hydrophobic residues show improved fits and are likely preferentially accommodated by the NBD1 and NBD2 pore loops (Fig. 4). The NBD1 is predicted to favor strong interactions with aromatic residues at alternating positions due to coordination by the KYR motif. This analysis agrees with pre-vious studies identifying ClpB binds preferentially to peptides containing aromatic residues[16] and indicates that the channel can readily accommodate sequences containing bulky side chains. Additionally, from our results we predict that sequences which are low complexity or contain repeats of charged residues or residues with small sidechains (such as Gly, Ala, and Pro) would be unfavorable and potentially inhibitory to translocation by ClpB. Indeed, diversity in the substrate sequence is favored by the proteasome[51] and Gly–Ala repeat sequences impair unfolding and trigger release, a mechanism by which the Epstein Barr virus protein EBNA1 evades proteasomal degradation[52,53].

Surprisingly, we identify the substrate entrance channel com-prises of an NTD trimer from alternating protomers (P1–P3–P5). In the WT Hsp104:casein structure density for all six NTDs was identified[6], while in the ClpB[BAP]:casein structure a trimeric NTD arrangement was identified for one of the classes, supporting the arrangement observed here[5]. Notably, two isoforms, a full-length (ClpB95) and an NTD-minus ClpB (ClpB80), are present in *E. coli* and have been shown to form heteroligomers and function synergistically in disaggregation[54]. Therefore, ClpB may have evolved to function optimally as a heterohexamer with an NTD ring that consists of primarily three protomers which dynamically interchange to bind substrate during cycles of NBD-driven ATP hydrolysis and translocation. From focused classification of the ClpB NTD ring we identify the polypeptide substrate is bound by the NTD trimer, extending from the AAA+ channel at the P5-pore loop position to contact the known substrate-binding hydrophobic groove[36] in protomers P3 and P5 (Fig. 5c), reveal-ing how the NTDs contribute directly to substrate interactions.

Based on our analysis of the Pre and Post states, we propose a rotary cycle, similar to models proposed for Hsp104 (ref.[6]) and in other studies[38,39]. By comparison of the Pre and Post con-formations, we identify the NBD1 of the protomer unbound to substrate (P6) shifts upward along the substrate axis, on-path to the next contact site (Fig. 6c), while the adjacent P1-NBD1

disengages from substrate and converts to an ADP state, supporting a hydrolysis-driven substrate release mechanism. Notably, a complete spiral arrangement with six protomers bound to substrate, identified for Hsp104 (ref. [6]) was not observed in any classes, indicating ClpB is likely tuned differently. Finally, we identify substantial conformational changes between the NBDs that also coincide with changes in nucleotide state and substrate release at the seam. ATP hydrolysis by NBD1 and NBD2 is cooperative and allosterically controlled[55,56]. Thus, these conformational changes support ATP hydrolysis-driven coupling between the domains that may promote substrate release at lower contact sites. Additionally, these changes could enable directional translocation by switching between high-affinity and low-affinity protomer interface conformations such that ATP-re-binding favors engagement of the clockwise, ATP-state protomer and the upper substrate position. Together these changes support a processive translocation mechanism involving sequential hydrolysis-driven steps that pull, stabilize, and release substrate during disaggregation. Conversely, changes in the substrate sequence or aggregation state may alter the gripping capacity of the pore loops, triggering stochastic hydrolysis or non-processive translocation events, which have been previously described[57,58].

## Methods

**Purification and analysis of ClpB variants.** *Escherichia coli* ClpB variants: ClpB$^{K476C}$, ClpB$^{K250E}$, ClpB$^{R252E}$ ClpB$^{E254K}$, ClpB$^{E256R}$, ClpB$^{E639K}$, ClpB$^{K640A}$, ClpB$^{K640E}$, ClpB$^{H641A}$, and ClpB$^{H641E}$ were generated using the QuikChange Site Directed Mutagenesis kit (Agilent) (Supplementary Table 4). ClpB protein was expressed with a C-terminal His$_6$-tag constructs from the pDS56/RBSII plasmid. Freshly transformed M15 cells (Qiagen) were inoculated in 2XYT media with 100 μg/ml Ampicillin and grown at 37 °C to OD$_{600nm}$ = ~0.7–0.8. Cells were induced for ~12 h at 15 °C with 1 mM IPTG. Cells were lysed by sonication in buffer containing 40 mM HEPES pH 7.4, 500 mM NaCl, 20 mM imidazole, 10% glycerol (v/v), and 2 mM β-mercaptoethanol with 5 μM pepstatin A, cOmplete Protease Inhibitor Cocktail (Roche), and lysozyme. The lysate was clarified by centrifugation (16,000 × g, 20 min, 4 °C) and incubated with Ni-NTA (GE Healthcare) for 3 h at 4 °C. The protein was eluted with elution buffer (40 mM HEPES pH 7.4, 500 mM NaCl, 500 mM imidazole, 10% glycerol (v/v), and 2 mM β-mercaptoethanol). Following dialysis into the lysis buffer, purification was performed by chromatography using a HisTrap$^{TM}$ HP column to remove additional contaminants. Purity was verified by SDS-PAGE and the fractions were combined and concentrated into a storage buffer (40 mM HEPES pH 7.4, 500 mM KCl, 20 mM MgCl$_2$, 20% glycerol (v/v), and 2 mM β-mercaptoethanol).

For ATPase assays, ClpB (0.25 μM monomer) was equilibrated in luciferase refolding buffer (LRB, 25 mM HEPES-KOH, pH 7.4, 150 mM KAOc, 10 mM MgAOc, 10 mM DTT) for 15 min on ice and then incubated for 5 min at 25 °C in the presence of ATP (1 mM) plus or minus casein (10 μM). ATPase activity was assessed by the release of inorganic phosphate, determined by malachite green phosphate detection (Innova). Background hydrolysis was determined at time zero and subtracted. For luciferase disaggregation assays, aggregated luciferase was first generated by incubating firefly luciferase (50 μM) in LRB plus 8 M urea at 30 °C for 30 min. The sample was then rapidly diluted 100-fold into LRB and frozen at −80 °C. Aggregated luciferase (100 nM) was incubated with ClpB or ClpB$^{K476C}$ (1 μM monomer), DnaK (1 μM), DnaJ (0.2 μM), and GrpE (0.1 μM), plus ATP regeneration system (5 mM ATP, 1 mM creatine phosphate, 0.25 μM creatine kinase) or 5 mM ATPγS for 90 min at 25 °C. Luciferase activity was assessed with a luciferase assay system (Promega). Recovered luminescence was monitored using a Tecan Infinite M1000 plate reader.

GREMLIN coevolution analysis and logo plot (Supplementary Fig. 2c) were performed using OPENSEQ.org web server supported by David Baker's lab (http://gremlin.bakerlab.org/)[59], using *E. coli* ClpB as the primary sequence. The HHblits was used to generate the diversity multiple sequence alignment (MSA) and 6177 sequences were analyzed. An *E*-value of $1E^{-10}$ and interactions of four were chosen to control the MSA generation. Filter MSA parameters were set to remove sequences that did not cover at least 75% of query. After coverage filter, positions in the alignment that have 75% of gaps were removed.

**Cryo-EM data collection and processing.** For substrate-bound complex formation, ClpB protein (150 μM) was incubated with ATPγS (5 mM) in the presence of FITC-casein (150 μm) (#C0528; Sigma) in buffer containing: 40 mM HEPES (pH = 7.5), 40 mM KCl, 10 mM MgCl$_2$, 1 mM DTT. Size exclusion chromatography (SEC) analysis and purification in the same buffer using a Superose 6 Increase 3.2/300 column (GE Healthcare) and fractions were collected and analyzed by SDS-PAGE. The ClpB-casein eluted at 1.5 mL (Supplementary Fig. 1a). Following SEC fractionation the samples were diluted to ~1.5 mg/mL, applied (1.5 μl) to glow-

discharged holey carbon grids (R 1.2/1.3; Quantifoil), plunge-frozen using a vitrobot (Thermo Fischer Scientific), and imaged on a Titan Krios TEM operated at 300 keV (Thermo Fisher Scientific). Images were recorded on a K2 Summit detector (Gatan Inc.) in super-resolution mode at 48,450×, corresponding to a calibrated pixel size of 1.032 Å/pixel after 2× Fourier-binning. Data were collected using Serial-EM[60] with 8s exposures at 200 ms/frame, and a total electron dose of 56$e−$ per micrograph in 40 frames. Whole-frame drift correction was performed with MotionCor2 (ref. [61]) and the first two frames were removed. Micrographs were CTF corrected using Gctf[62] and single particles were picked using Gautomatch (K. Zhang, http://www.mrc-lmb.cam.ac.uk/kzhang/Gautomatch/) and totaled 778,521 from 8499 micrographs. Subsequent processing was performed in cryoSPARC[63]. Contamination and junk particles were removed following 2D classification analysis and amounted to ~8% of the dataset. An ab-initio model was created from the original particle set, and used for subsequent classification and refinement steps. The initial homogeneous refinement of all 700 K particles yielded a 2.9 Å overall resolution structure by the "gold standard" 0.143 Fourier shell correlation (FSC) criterion (Supplementary Fig. 1d).

The Pre and Post states were determined following a 20-class 3D classification using Relion 2.1 (ref. [64]) with the ClpB$^{K476C}$ map low passed filtered to 60 Å as an initial model (Supplementary Fig. 6a). The Pre-state (239,000 particles) was overall identical to the map refined using the total dataset, while the Post-state (213,000 particles) exhibited changes in protomers P1 and P6. The remaining classes, totaling ~248,000 particles were not well-resolved and excluded. Following 3D refinement the Pre and Post states refined to 3.4 and 3.7 Å, respectively. Separate focus classification[65] analyses were performed to better-resolve the P1–P6 seam protomers and NTD ring. The approximate center coordinates of P1–P6 protomers and the NTDs were determined in Chimera[66] (Supplementary Figs. 5a and 6b). The particles were re-centered and extracted using pyem (D. Asarnow, https://github.com/asarnow/pyem) and then 3D classification was performed without image alignment, using a reference map and mask corresponding to the two front protomers or NTDs and adjusted to the new center. Focus classification of the P1–P6 protomers resolved the same Pre and Post conformations but showed improvement in the density for the protomers. 3D refinement of the individual classes was performed and the final maps included ~220,000 and ~91,000 particles for the Pre- and Post-states. For the NTD focus classification, classes with improved NTD definition were selected for 3D refinement, and included ~93,000 particles. During 3D refinement the initial angular sampling value was set to the same value as the local search value. The WT ClpB dataset was recorded in counting mode at 50,000×, corresponding to 1.0 Å/pixel using Leginon[67] with 8 s exposures at 200 ms/frame, and a total electron dose of 52$e−$ per micrograph in 40 frames. 3D classification and refinement was carried out similarly as above with ~55,000 particles in the final refinement out of ~110,000 classified in 3D, and a resolution of 4.1 Å for the final map. All of the final maps underwent "Post-processing" procedures within RELION[68] or cyroSPARC[63] to sharpen and determine FSC estimation. ResMap[69] was used to estimate the local resolution.

**Molecular modeling.** An initial model was determined by rigid body fitting the protomers from the previous ClpB structure (pdb:5ofo)[5] using UCSF Chimera's[66] *Fit in Map* function. The D1′ loop residues and substrate (as a poly-Ala chain) were built using Coot[70]. Using Phenix[71], the docked model underwent one round of simulated_annealing followed by 10 cycles of real-space refinement that consisted of minimization_global, rigid_body, adp, secondary structure restraints, and non-crystallographic symmetry (NCS) restraints. The resulting model then underwent manual real-space refinement in Coot[70]. Following Molprobity validation, the model underwent another 10 cycles of real-space refinement in Phenix with minimization_global, rigid body, and adp. For the Post-state model, the NBDs of P1 and P6 were individually docked using UCSF Chimera[66] and refined in Phenix[71], as above.

For the casein sequence modeling, overlapping sequences from the four bovine isoforms, α-s1, α-s2, β, and κ (Sigma, #C0528) were modeled using Rosetta[47]. A backbone model was initially built manually and a "symmetrized" model was then generated by sampling four parameters corresponding to the phi/psi backbone angles of residues 1 and 2 of the two-residue repeat unit, and extending it over the 13 resolved residues in both the NBD1 and NBD2 rings. The phi and psi angles were sampled at two-degree increments, each evaluated against the density in both NBD1 and NBD2 rings, and a final model with phi$_1$ = −92, psi$_1$ = 118, phi$_2$ = −102, and psi$_2$ = 110 was selected. Symmetrizing the model in this manner was used to avoid overfitting the backbone model to the modest-resolution density.

Next, using the *partial_thread* tool in Rosetta, 1604 models of the casein sequence were threaded onto this backbone, 802 into the NBD1, and 802 into the NBD2 ring. Each of these models was refined using Rosetta's *relax* protocol with an additional term enforcing agreement to density at a modest weight (ensuring the backbone of the peptide would not refine far from the starting point). Rosetta's *relax* allows refinement of both backbone and sidechain residues[72]; in this case, relax was restricted to only allow movement of the casein peptides themselves and the pore loops of NBD1 and NBD2, for a total of about 112 moving residues in each refinement. NBD1- and NBD2-interacting peptides were modeled separately.

For each of the 1604 threaded models, five independent refinement trajectories were carried out, and the lowest-energy model was selected for analysis. From these sequence-energy pairs, a profile was constructed by computing—for each amino

acid at each position—the average energy over all sequences with the corresponding amino acid at the corresponding position. The N- and C-terminal residues were excluded from this analysis. To make energy comparisons valid between diverse sequences, a sequence-specific reference weight was used in Rosetta to capture unfolded state energetics. Finally, a "null-background" model was constructed, in which every residue except one was modeled as alanine: at this residue, all 20 amino acid identities were generated. This led to a total of 440 models (11 positions × 20 amino acid identities × 2 rings), which were refined in the same manner as the casein peptides. For these models, a profile was constructed by comparing the energy of each position-residue pair to the average energy of all evaluated peptides. These profiles showed similar trends to the casein-threaded profiles. Sequence logo plot was generated by weblogo 2.8 (ref. [73]).

## Data availability

ClpB[K476C] cryo-EM maps and atomic coordinates have been deposited in the EMDB and PDB with accession codes EMDB-20004 and PDB-6OAX for the Pre state, EMDB-20005 and PDB-6OAY for the Post state, EMDB-20049 and PDB-6OG1 for Pre state focus class, EMDB-20050 and PDB-6OG2 for the Post state focus class, and EMDB-20051 and PDB-6OG3 for the NTD focus class. The source data underlying Figs. 1a, 2f, and 3e and Supplementary Figs. 1d, and 2b are provided as a Source Data file. Other data are available from the corresponding author upon reasonable request.

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

## Acknowledgements

We thank Korrie Mack, Zachary March, and Ryan Cupo for feedback on the manuscript. We thank the UCSF BACEM Facility for assistance with data collection. This work was supported by an Alzheimer's Association Research Fellowship (to J.B.L.), an NSF Graduate Research Fellowship DGE-0822 (to L.M.C.), a GAANN fellowship (to A.N.R.), NIH grants R01GM099836 (to J.S), and R01GM110001 (to D.R.S.).

## Author contributions

A.N.R. prepared grids, collected and processed cryo-EM data, and built and refined models. J.B.L. purified protein, and performed mutagenesis and activity and binding assays. S.N.G. performed initial cryo-EM analysis. E.T. performed microscope alignments, sample loading, and assistance in data collection. S.M.B. and L.M.C. performed mutagenesis and protein purification. F.D.M. performed substrate modeling experiments. All authors edited and read the manuscript. J.S and D.R.S. performed experimental design, data analysis, and manuscript preparation.

## Additional information

**Competing interests:** The authors declare no competing interests.

