## [Peer Review File · Nature Communications]

Reviewers' comments:

Reviewer #1 (Remarks to the Author):

This manuscript describes a cryo-electron microscopy analysis of the complex of a bacterial ATP-dependent disaggregase ClpB with its pseudo-substrate casein. The structural reconstruction of the ClpB-casein complex reached ~3Å resolution and gives an unprecedented wealth of near atomic-level detail on how ClpB interacts with its substrate, with important implications for understanding the mechanism of ClpB-induced disaggregation of aggregated proteins. This is a high impact paper which can be recommended for publication in Nature Communications after a revision (see below) and with the following caveat:

A very similar study has just been published by another group (Yu et al., ATP hydrolysis-coupled peptide translocation mechanism of Mycobacterium tuberculosis ClpB) in PNAS with the publication date Sep 26, 2018 (sent for review on June 21). The PNAS paper and the present manuscript contain an almost identical experimental design with ATP γ S used to stabilize the casein-ClpB complex and a high-resolution cryo-EM analysis. Some of the most interesting observations from the present paper, like the significance of basic residues flanking the Tyr in the ClpB pore loop in NBD1, were also discovered in the PNAS paper. The present paper contains novel structural analysis of the interaction between the N-terminal domain of ClpB and casein, which is not included in the PNAS paper from the competing group. The present manuscript was sent for review on Sep 27, only a day after the PNAS paper was published, so the two papers are clearly independent and simultaneous discoveries, and both deserve publication in high-quality journals. However, I believe that the Authors of this manuscript should include a comment on the PNAS paper and perhaps compare its findings with their own results.

Specific comments:

1. Some references in the text are used quite arbitrarily, which can be misleading for the readers; for example, ref. 24 is outdated because it presumed a C-terminal binding site in ClpB and did not describe conformation of the middle domain (ref. 27 should be used instead). Also ref. 31 shows an alternative (and controversial) structural model of ClpB/Hsp104 and should not be used to support the location of the amino-terminal domains for this study.
2. Rationale for using the hyperactive variant ClpB K476C is not clear in the Introduction. In the text, the authors argue that K476C is structurally equivalent to wt ClpB, but the structural resolution for the latter was worse. Perhaps, this should be stated out front.
3. Line 135: The ClpB K476C activity with ATP γ S is not robust, as the Authors state, it is rather marginal (see Fig. 1A).
4. Fig. 1A. This is an important experiment to demonstrate that a substrate can be threaded by ClpB in the absence of an efficient ATP hydrolysis. However, the "substrate" used in the EM analysis is casein, whereas the substrate used in the biochemical assay is an inactivated luciferase. There have been established assays for threading of FITC-casein by ClpB and such an assay should be included in this paper to provide a biochemical support for imaging the K476C ClpB-casein complexes with ATP γ S. Moreover, reactivation of luciferase in the presence of ATP γ S requires the DnaK system of co-chaperones, which are not included in the structural analysis. So, the data in Fig. 1A do not support the experimental design used in this study. Eliminating DnaK from this experiment is also important because the K476C mutation is within the DnaK-interacting domain of ClpB, so an assay with DnaK can be difficult to extrapolate into a DnaK-independent ClpB-substrate interaction.
5. Line 148, Fig. 1D. "This density is attributed to a 27-residue unfolded strand of the casein substrate". ClpB structure without casein should be shown for comparison, together with a difference map.
6. Line 238: which casein isoform was used in this study (alpha, beta, kappa)? It would be useful to include the casein sequence in the supplement.
7. Line 241: the sequences of casein peptides modeled as bound to ClpB show a strong similarity

to the hydrophobic ssra tag, which is recognized by ClpA and ClpX, but not by ClpB (Hinnerwisch et al, Cell vol 121, p. 1029, 2005). It seems that the casein mode of binding to ClpB may not reflect the physiological mechanism for physiological substrates. The Authors should comment on this limitation.

8. Par. from line 238: Where are the identified peptides located in the sequence of casein? Are they close to either the N- or C-terminus? Why are the remaining regions of casein invisible in the imaging analysis? Why did the Authors not try to identify the bound peptides by analyzing the protection of casein by ClpB during a limited proteolysis? An experimental approach could be more accurate than peptide modeling.

9. Fig. 5. Which structural data set for ClpB NTD was used for modeling? This reviewer attempted to compare the NTD structure shown in Fig. 5C with the PDB set 1KHY and found that the helices A1 and A6 could have been mislabeled in Fig. 5C, left (i.e. A6 is on top, A1 on bottom). Please, verify the images in Fig. 5c and the helix labels.

10. An apparent discrepancy in Fig. 5c brings another question: have the Authors considered how NTD is connected with NBD1 during fitting of the NTD structure to the EM envelope? Since NTD is purely helical and quite globular, how accurately can the Authors' model discriminate between different orientations of NTD? Indeed, it appears that the NTD images in the left and right panels of Fig. 5c were not obtained by a rotation around a vertical axis, but through an "upside-down" flip. The C-terminus of P5 NTD points down in the left panel, but it points up in P3 NTD (right panel). Are both the NTD structures shown in Fig. 5c compatible with their connection to NBD1?

11. There is an experimental evidence that a triad of residues: T7 at the N-terminus of A1, D103 at the C-terminus of A6, and E109 are involved in interactions with aggregated substrates of ClpB (Liu et al. JMB vol 321, p. 111, 2002; Barnett et al., JBC vol. 280, p. 34940, 2005). How far is that residue triad from the casein chain in P1, P3, and P5 NTDs?

Reviewer #2 (Remarks to the Author):

Summary:

The authors present a cryo-EM structure of the hyperactive ClpBK476C mutant bound to FITC-casein and in the presence of ATP gamma S. While previous structures of ClpB exist, the authors set out to elucidate the structure of an active, substrate-bound ClpB at higher resolution to shed light on the translocation mechanism. The authors show that this mutant has higher ATPase activity and has disaggregase activity in vitro. The authors demonstrate that five promoters make direct contacts with the substrate, while the last promoter is more flexible and does not make direct substrate contacts, similar to other substrate-bound AAA+ structures (such as those from this lab).

The structure presented here is at high enough resolution that the authors can identify specific pore-loop interactions that are working to stabilize the substrate. For NBD1, the authors show that the conserved tyrosine in pore loop 1 and the two flanking basic residues form a clamp around the substrate. The authors identify a novel cross-loop interaction where the two basic residues flanking the tyrosine form salt bridges across the pore loop. The authors validated the importance of these salt bridges by mutating the basic residues and testing disaggregation. Consistent with the importance of these residues in ClpB disaggregase activity, the mutants were defective in luciferase disaggregation. By contrast, the NBD2 pore loops do not seem to make cross-loop interactions and substrate interactions are primarily made by the canonical Ar/Φ motif. Furthermore, the authors show that two residues in the D2' loop make additional interactions with the substrate as it exits the channel. The authors propose a model in which the two pore loops interact with the substrate in distinct ways. The resolution of the substrate was also sufficiently high that the authors were able to determine the nature of the residues that can be accommodated by ClpB and validate the expectation that aromatic and hydrophobic residues are more favorable.

An important novelty of this work is that prior studies were unable to resolve the structure of the N-Terminal Domains due to their flexibility, leaving a gap in our understanding of the nature of the interactions at the entrance of the channel. In this study, the authors were able to use focused refinement to identify that the NTD ring is made up of three domains which make contacts with alternating protomers as well as direct contacts with the substrate. Importantly, substrates seem to contact helical regions previously implicated in biochemical studies to be important for ClpB function.

The authors conclude with identifying two distinct conformations that shed light on the translocation mechanism and showing that the distance between NBD1 and NBD2 increases at the lower promoters. Taken together, the authors propose that translocation by ClpB involves a rotary mechanism where ATP hydrolysis is coupled to substrate release and further ATP binding leads to substrate binding at the next contact point.

Strengths:

Although previous structures of ClpB have been published, this work elucidates novel interactions and proposes a well-thought out mechanism. For instance, the authors show that for NBD1, residues K250 and R252 are important for disaggregase activity, which strengthens their model. An additional novel aspect of this work is the resolution of the NTD, showing that it is potentially contributes to substrate translocation directly.

Weaknesses:

The model proposed is elegant and logical. However, the authors fail to test the proposed mechanism fully. For example:

1. The authors test the importance of the proposed cross-loop interactions in NBD1, but they do not test the importance of some of the proposed interactions in NBD2. The authors propose that residues E639 and H641 interact directly with the substrate as it is exiting the channel and likely form additional stabilizing interactions that are important for substrate release. However, the authors do not mutate these residues and test disaggregation activity as they did for the cross-loop interactions. It would ideal to see the impact of these mutations as well.
2. The authors suggest that salt bridges formed by K250 (to E254) and R252 (to E256). The single mutations clearly show the importance of these basic residues, but complementing these charge inversions to restore the salt bridge would be a powerful validation of their model. Does the E254K or E356R restore activity in the K250E or R252E background?
3. The authors suggest that the NTD ring plays a direct role in substrate binding and transfer to the NBD domains based on their structure and on prior work (for example Rosezweig, et al) showing an important role for the NTD in substrate binding and activity. The surprising proposal here is the model that substrate recognition and processing uses only three NTDs (as a trimeric ring) as part of the disaggregation cycle, which the authors suggest is consistent with the enhanced activity seen in prior work with mixtures of ClpB isoforms. However, experimentally testing this model directly would be ideal. For example, do mutations on the surfaces of the NTDs that are not involved in the trimeric ring interface have no effect on substrate translocation? Can mutations that disrupt this trimeric structure (without effecting the A6/A1 helices) alter substrate translocation/recognition?

Minor concerns:

Figures 1-2. The activity data is described as coming from two reactions (n=2), it would be better to simply show these two data points as a dot plot with the mean shown given the very small sample size.

Reviewer #3 (Remarks to the Author):

The manuscript of Rizo and colleagues describes cryo EM structures of the hyperactive ClpBK467C variant. Aim of the study was to elucidate conserved mechanisms of disaggregation between related AAA+ Hsp100 chaperones. The authors present ClpB structures bound to the substrate casein and the slowly hydrolysable nucleotide analogue ATPS in the so-called "Pre" and "Post" states. At a resolution of 2.9Å the substrate density and pore loops are well resolved. The most convincing finding is that salt bridges between charged residues adjacent to the pore loop tyr in NBD1 stabilise the loop and are important for ClpB disaggregase activity. It is also interesting that the substrate is bound by non-neighbouring N-terminal domains and can be traced all the way from the N-terminal domains to the C-terminal exit at the ClpB pore. However, the hypothesis that NBD1 and NBD2 specifically interact with preferred substrate sequences is not very convincing. The interpretation that increased separation between NBD1 and NBD2 triggers ATP hydrolysis is also questionable. The manuscript could be shortened and contains several ambiguous statements as well as errors. Overall, the manuscript contains many original findings. Although the structure of substrate bound ClpB is not novel, the resolution of the EM structure presented here is significantly better than the published structures. The direct comparison between ClpB and Hsp104 structures solved by the same group contributes to resolving dispute about structural differences between Hsp100 proteins in the field.

Major points:

- The extracted casein density in Figure 1d merely shows any side chain densities. This is expected as there are no reports of Hsp100 substrate preference and the casein density will thus be an averaged density over all possible substrate processing steps. Biochemically, no such preference or substrate stalling has been shown. It is surprising that the authors attempt to fit 1604 models of casein peptides into the density. It is very difficult to understand how the fits were weighted and thus how the energy scores were obtained. It does not seem surprising that the highest scoring peptides all have many alanines and small hydrophobic side chains, as these would probably best fit the by and large side-chain-free density of casein. Also, it is expected that the tyrosine residue in the substrate binding loop will preferably interact with substrate via stacking. stacking can be observed in many protein-protein and protein-cofactor interactions and is well characterised. Besides responding to the points above the authors should address the following questions: Which scoring functions are used for determining the energy scores of substrate interaction? Is there any indication that the modelled sequences are preferred by the enzyme in vitro? What effect would the passage of a favourable binding sequence through ClpB have? Would the peptide bind stronger to the pore loops and passage would be slower?
- Suppl. Figure 5c: The images are difficult to comprehend. The densities for nucleotide and arginine finger are almost impossible to make out between the secondary structures shown. Often the arginine finger is not even drawn into the image (post p3-p5 NBD1 and NBD2, post p2 NBD1, pre p1 NBD1) and it is unclear which density corresponds to the arginine that is supposed to be in contact with the nucleotide in these states. The density for the nucleotide is very weak in many states (post p1 NBD1 and NBD2, post p6 NBD1, post p2 NBD1 and NBD2, post p3-p5 NBD1 and NBD2). The authors should show isolated densities for the nucleotides and arginines and explain how they deduce which nucleotide is occupying a particular binding pocket. The images in this figure don't seem to be on the same scale.
- The interpretation of NBD1-NBD2 separation in the last paragraph of the discussion does not add up very well. In line 470f the authors argue that the separation could function as a pulling mechanism. How does this work if the protomers in question (p1, and particularly p6) are not even connected to the substrate? The increase in distance in p1 is negligible (2 Å). The strong increase is found in p6, which is not in contact with casein. The authors further claim that the separation between NBD1 and NBD2 increases, the lower the position of the protomer along the substrate. This is supposed to add strain to the substrate-protomer interaction and trigger ATP hydrolysis. P1 and p6 however, are already in the post-hydrolysis, ADP bound state, lacking arginine finger interactions crucial for hydrolysis. How can hydrolysis occur without arg-finger interaction? Also, p6 shows the strongest domain separation but is not at the lowest position in the hexamer. From their own data, one would conclude that domain separation is following ATP hydrolysis rather than triggering it. The authors should revise this section.
- The manuscript lacks a clear comparison between the ClpBK467C (ClpBwt) and ClpBBAP variants. The authors should discuss why the ClpBK467C variants miss density for the middle domain, while it can be seen in the substrate bound ClpBBAP variant. Although some similarities between the maps are made clear (hand, overall layout, no extended state) it remains unclear

whether the authors observe clear differences to the already published structures.

Minor point:

- Line 114: the authors should clarify what they mean by the "stability was problematic under cryo conditions". Was complex formation poor, did the protein get degraded or did the protein aggregate?
- Line 116: an explanation of what "closed" state means is required rather than merely putting it in quotation marks.
- Line 333: this can only be the P6 Arg finger position
- Suppl. Figure 1h: The FSCs are truncated at $\sim 4 \text{ \AA}$. They need to be shown up to Nyquist frequency. The current diagram makes it impossible to assess whether the FSC resolution estimate is inflated. The FSC needs to drop to zero at high frequencies.
- Figure 1b and 1c lack a scale bar

(Spelling) errors:

- Line 211 should presumably read "substrate stabilizing contacts"
- Line 338: should presumably read "from the clockwise protomer for both..."
- Line 346: assuming that the authors are talking about the seam interface it would be better to consistently refer to it as seam rather than spiral interface.
- Line 362: should read "show similar changes around the hexamer"
- Line 387: delete "for the"
- Line 409: should read less severe
- Line 424: should read would exit the translocation channel
- Line 455: should read resulting in a processive translocation step
- Line 535,548,553,558: the image processing method is called "focussed classification" not focus classification
- Line 569 should read counting mode
- Line 579 should read total particle set after 2D classification

Response to Reviewers' comments.

Overall, we are grateful to the reviewers for taking the time to provide important and insightful feedback on this manuscript. All three reviewers commented positively about the work, indicating: “this is a high impact paper which can be recommended for publication in Nature Communications” (Reviewer 1), “this work elucidates novel interactions and proposes a well-thought out mechanism” (Reviewer 2), and “overall, the manuscript contains many original findings” and “contributes to resolving dispute about structural differences between Hsp100 proteins in the field” (Reviewer 3).

Additionally, the reviewers each identify a number of issues and clarifications which we have specifically addressed. In particular, we have now included additional mutagenesis data, including disaggregation and substrate binding analysis, that provides further support of a critical functional role for the NBD1 cross pore-loop interactions and NBD2' loop contacts we identify in the structure. These data have been added to Figures 2f and 3e. Responses to specific points from each reviewer are described below. A number of responses are cited, with references at the end.

Reviewer #1

Reviewer 1 brings up the recent publication of the ClpB structure from *M. tuberculosis* (Yu, Lupoli et al. 2018) and suggests that we provide comment about it now that it is published.

We have now referenced this publication and added the following point to the Discussion section:

“Notably, the ~3.9-Å structures of ClpB from *Mycobacterium tuberculosis*, published during the review of this manuscript, are consistent with the hexamer architecture and stepwise translocation mechanism proposed here, however there are some differences in the position and conformation of the seam protomers in the two states compared to the structures presented here. Additionally, the NTD-trimer arrangement with the bound polypeptide substrate was not resolved in this previous study.”

1. Some references in the text are used quite arbitrarily, which can be misleading for the readers; for example, ref. 24 is outdated because it presumed a C-terminal binding site in ClpB and did not describe conformation of the middle domain (ref. 27 should be used instead). Also ref. 31 shows an alternative (and controversial) structural model of ClpB/Hsp104 and should not be used to support the location of the amino-terminal domains for this study.

We thank the reviewer for making these suggestions and have corrected the indicated references (using Ref. 27 in place of 24 and removing 31) and gone through the manuscript to remove other citations that may be considered less pertinent.

2. Rationale for using the hyperactive variant ClpB K476C is not clear in the Introduction. In the text, the authors argue that K476C is structurally equivalent to wt ClpB, but the structural resolution for the latter was worse. Perhaps, this should be

stated out front.

We have made the following changes to the Introduction: "... we sought to determine high-resolution cryo-EM structures of a substrate-bound ClpB complex that is active for ATP hydrolysis and polypeptide translocation. The ClpB^{K476C} MD variant was chosen due to its established hyperactive function(Oguchi, Kummer et al. 2012) and stability in substrate binding."

3. Line 135: The ClpB K476C activity with ATPgammaS is not robust, as the Authors state, it is rather marginal (see Fig. 1A).

We agree with the reviewer that the activity with ATPgammaS is not "robust" and have removed the adjective – this was meant to reference the activity with ATP, however the sentence was not clear. **The sentence now states:** "The ClpB^{K476C} casein-bound complex, incubated with ATPyS and purified by SEC, was targeted for high-resolution structure determination by cryo-EM." Indeed, upon further trials we found that like WT ClpB, ClpB^{K476C} lacked substantial disaggregase activity in the presence of ATPgammaS (Figure 1a).

4. Fig. 1A. This is an important experiment to demonstrate that a substrate can be threaded by ClpB in the absence of an efficient ATP hydrolysis. However, the "substrate" used in the EM analysis is casein, whereas the substrate used in the biochemical assay is an inactivated luciferase. There have been established assays for threading of FITC-casein by ClpB and such an assay should be included in this paper to provide a biochemical support for imaging the K476C ClpB-casein complexes with ATPgammaS. Moreover, reactivation of luciferase in the presence of ATPgammaS requires the DnaK system of co-chaperones, which are not included in the structural analysis. So, the data in Fig. 1A do not support the experimental design used in this study. Eliminating DnaK from this experiment is also important because the K476C mutation is within the DnaK-interacting domain of ClpB, so an assay with DnaK can be difficult to extrapolate into a DnaK-independent ClpB-substrate interaction.

We have now added a fluorescence polarization experiment that shows ClpB K476C binds casein in the presence of ATPyS with a K_d of 120 nM and is equivalent to WT under the same conditions. **These data have been added to Supplemental Fig. 1d and discussed in the Results and establish that the ClpB complex is bound to casein under the conditions used for cryo-EM, in which the protein concentrations are ~1-2 μM.** Moreover, it was previously established that ClpB^{K476C} (in the context of BAP) can thread casein through its central channel and into ClpP for degradation (see Supplementary Figure 6b in (Oguchi, Kummer et al. 2012)). Finally, other data in the literature have established that ClpB can thread soluble unfolded substrates, like casein, in the absence of efficient ATP hydrolysis (e.g. see (Nakazaki and Watanabe 2014)). We cite these papers on p. 6.

5. Line 148, Fig. 1D. "This density is attributed to a 27-residue unfolded strand of the casein substrate". ClpB structure without casein should be shown for comparison, together with a difference map.

A difference map between a substrate-bound and a substrate-free structure is not expected to properly resolve the substrate density because the hexamers likely adopt

different conformations. A substrate-free ClpBK476C complex is expected to adopt an “open” state arrangement similar to what we have previously characterized for Hsp104 (Yokom et al., *NSMB*, 2016 and Gates et al., *Science* 2017) and was also observed in the recently published study of ClpB from *M. Tuberculosis* (Yu et al., *PNAS*, 2018). These substrate free structures are substantially different than the closed, substrate bound conformation (e.g. the spiral adopts a left-handed twist compared to the right-handed twist of the substrate-bound state and the channel is more open), thus the hexamers in the two states would not align properly and the resulting difference map would be unable to resolve the substrate density.

Additionally, we have made initial attempts to determine structures of ClpBK476C in the absence of substrate, however these have been unsuccessful. We have tested conditions with AMPPNP, ADP and ATP γ S (but without casein) and found that the hexamer is relatively unstable under cryo-EM conditions. We have provided micrograph images of ClpBK476C-ATP γ S with and without casein that illustrate the instability (see below). These data were not included in the manuscript. However, given that open spiral conformation is now well-established as a substrate-free state by the previous studies we did not pursue it further for this study.

Comparison of ClpB-ATP γ S with and without casein showing improved hexamer stability in the presence of the casein substrate. Cryo-EM micrographs of samples: ClpB_K476C + ATP γ S (left) and ClpB_K476C + ATP γ S + Casein (right). Individual particles are highlighted with red circle.

Finally, for additional clarification we determined a difference map between the model of the AAA+ domains alone and the final map. For generating this “substrate-free” map all the atoms in the NBD domains were included, thus the density remaining in the difference map would correspond to molecules that are not a part of the ClpB hexamer. Indeed, this difference map shows an 80 angstrom-length of density across the channel that is identical to what we have modeled as substrate (Fig. 1d). The difference map is shown below.

Difference map (yellow, experimental map – model) of the AAA+ domains shown at 4 sigma. The primary pore loops are shown as ribbons around the density.

6. Line 238: which casein isoform was used in this study (alpha, beta, kappa)? It would be useful to include the casein sequence in the supplement.

The FITC-labeled casein used in this study (purchased from Sigma, cat. # C0528) is from bovine milk and is presumed to contain a mix of all four isoforms: alpha-s1, alpha-s2, beta, and kappa. **This information has been added to the Results and Methods sections. Sequences for all isoforms were used in the molecular modeling studies (Figure 4) and included in Table 2.**

7. Line 241: the sequences of casein peptides modeled as bound to ClpB show a strong similarity to the hydrophobic ssra tag, which is recognized by ClpA and ClpX, but not by ClpB (Hinnerwisch et al, Cell vol 121, p. 1029, 2005). It seems that the casein mode of binding to ClpB may not reflect the physiological mechanism for physiological substrates. The Authors should comment on this limitation.

We thank the reviewer for pointing this out. Upon review of (Hinnerwisch, Fenton et al. 2005), we found that ClpA and ClpB were not experimentally compared for binding to the ssra-tagged GFP in this study - only ClpA was investigated. Unfortunately, the authors reference unpublished data for their conclusion that ClpB does not recognize the ssra tag, thus we are unable provide a direct response based on this previous study. In other peptide binding studies, however, it has been shown that ClpB does bind the 11-amino acid ssrA peptide alone (with low-affinity) as well as longer peptides containing ssra-tagged peptides with affinities that are comparable to casein peptides (Li, Weaver et al. 2015). Additionally, ClpB has been shown to slowly unfold GFP-ssra constructs (Johnston, Miot et al. 2017) and mixtures of ATP and ATPyS have been shown to unleash the activity of ClpB, enabling interaction with RepA (Doyle, Shorter et al. 2007), which is normally ClpA/ClpX-specific, indicating plasticity of substrate

interactions is coupled to the hydrolysis mechanism. Thus, under certain conditions, ClpB can bind ssrA, and in general, substrate discrimination mechanisms in vivo may be more complex than the pore loop contacts we've modeled and involve additional factors or differences in kinetics. While it is well-established that ClpA and ClpX specifically recognize ssrA and RepA-tagged proteins (Keiler, Waller et al. 1996), additional work is needed, including high-resolution structures of substrate bound ClpA, in order to address a potential mechanism for substrate discrimination by ClpA compared to ClpB. Additionally, while these sequences have a commonality of being hydrophobic, there are differences between the top-scoring peptides we identify (VVTILALTLPF for NBD1 and ILACLVALALA for NBD2) and the ssra peptide (AANDENYALAA) that may provide discrimination, such as differences in steric constraints at specific positions (e.g. VV and IL compared to AA at the first two positions). Thus, we argue that our results are not contradictory to what is currently understood for sequence preferences of ClpA and ClpB.

8. Par. from line 238: Where are the identified peptides located in the sequence of casein? Are they close to either the N- or C-terminus?

For the NBD1 the lowest scoring (best fit) peptide (Fig. 4a) derives from the κ isoform, residues 8-18, and for the NBD2 the peptide is from the β isoform, residues 5-15. **This information has now been added to the Results.** While both these are located towards the N-terminus of casein, the position along the full sequence would be considered arbitrary and was not taken into account during the modeling. Indeed, several peptides fit into the NBD1 and NBD2 channels with similar scores and are located at various positions along the casein sequence. **Importantly, we have now added plots showing the full sweep of energies, plotted from the N- to C-termini for all casein peptides tested, relative to poly-Ala (Supplementary Fig 4).**

Why are the remaining regions of casein invisible in the imaging analysis?

The remaining regions of casein that are not bound in the channel are not observed in the cryo-EM map because these regions are expected to be highly flexible, unstructured and heterogenous relative to ClpB. These densities would then be blurred out during the cryo-EM processing due to averaging and not present in the reconstruction. The substrate would only be visible outside the channel if it was folded or perhaps bound homogeneously to either the NTD interface or the channel exit.

Why did the Authors not try to identify the bound peptides by analyzing the protection of casein by ClpB during a limited proteolysis? An experimental approach could be more accurate than peptide modeling.

A limited proteolysis/protection experiment followed by MS sequencing would be an excellent approach to determine which sequences were bound in the channel. However, the primary question we wanted to address in this modeling experiment was whether there are distinct NBD1 and NBD2 substrate specificities and what the per-residue interaction preferences are along the channel. Given the differences in channel architecture that we identify for the NBD1 and NBD2 (e.g. the KYR and YV pore loops) we wanted to see if these differences support different substrate interactions.

Additionally, in a previous study, substrate preferences for ClpB were determined using a peptide array (Schlieker, Weibezahn et al. 2004) and we wanted to identify whether the peptides that fit best into the channel by modeling have similar characteristics to what was previously determined biochemically. **This study is referenced and discussed in the Discussion section.**

9. Fig. 5. Which structural data set for ClpB NTD was used for modeling? This reviewer attempted to compare the NTD structure shown in Fig. 5C with the PDB set 1KHY and found that the helices A1 and A6 could have been mislabeled in Fig. 5C, left (i.e. A6 is on top, A1 on bottom). Please, verify the images in Fig. 5c and the helix labels. Indeed, these helices were mislabeled and we greatly appreciate that this was identified by the reviewer. We have made the appropriate change to Fig. 5c, with A1 being on bottom and A6 on top. PDB 1KHY was used for modeling the NTDs.

10. An apparent discrepancy in Fig. 5c brings another question: have the Authors considered how NTD is connected with NBD1 during fitting of the NTD structure to the EM envelope? Since NTD is purely helical and quite globular, how accurately can the Authors' model discriminate between different orientations of NTD?

The connecting density between the NTD and the NBD1 was identified for all three NTDs (Supplementary Fig. 5c) and was taken into consideration during the modeling. In fact, this was a key constraint for identifying which NTD belonged to which promoter and for verifying a proper fit, which was based on the proximity of the connecting density to the C-terminal residue of the NTD structure. **We have added the following clarification to the Methods:**

“The NTDs were modeled by manually docking the domains individually and then performing a rigid body fit with Chimera “Fit in Map” procedure. As verification of the rigid-body docking, the C-terminal strand of each NTD was confirmed to be adjacent the N-terminus of the NBD1 of the corresponding promoter and connecting density was identified that likely corresponds to the 19-residue strand between the domains, which is not present in previous crystal structures and, therefore, was unable to be modeled (Supplementary Fig. 5c).”

While the NTDs are globular, the resolution following the focused refinement improved such that tubular-shaped α -helical density was apparent at various threshold values, which greatly facilitates the fitting procedure (Supplementary Fig. 5c). During rigid body docking we tested a number of different starting positions and verified the fit both visually (looking for placement of α -helices in appropriate density and the position of the C-terminus) and by cross-correlation. **The cross-correlation values for each of the NTDs have now been included in Supplementary Fig. 5c.** With these approaches, we are confident of our model for the NTDs. It is worth noting, however, that the NTDs are dynamic, thus multiple conformations may be sampled during disaggregation cycles.

Indeed, it appears that the NTD images in the left and right panels of Fig. 5c were not obtained by a rotation around a vertical axis, but through an “upside-down” flip. The C-terminus of P5 NTD points down in the left panel, but it points up in P3 NTD (right

panel). Are both the NTD structures shown in Fig. 5c compatible with their connection to NBD1?

The discrepancy noted in Fig. 5c may be from removing the protomer P1 from the view. Also, residues 75-82 are not present in the 1KHY PDB, thus there are additional truncation points (located towards the top for P5) that may make it challenging to identify the C-terminus of the NTD. **We have modified the fig. 5c to focus on the NTDs from P3 and P5 that appear to contact the substrate. We have also noted the location of the C-terminal residue of the P3 and P5 NTDs (*).** The differences in the two views are an approximate 30deg counter-clockwise rotation about the Y-axis going from 5b to 5c. The images were not flipped around the X-axis. We removed the rotation note, however, because protomer P1 was removed from the view in 5c, and thus the image has a different composition.

11. There is an experimental evidence that a triad of residues: T7 at the N-terminus of A1, D103 at the C-terminus of A6, and E109 are involved in interactions with aggregated substrates of ClpB (Liu et al. JMB vol 321, p. 111, 2002; Barnett et al., JBC vol. 280, p. 34940, 2005). How far is that residue triad from the casein chain in P1, P3, and P5 NTDs?

We thank the reviewer for pointing out these previous studies. We have localized these residues in our model and have included an image here highlighting these positions (see below). However, these residues are reasonably far from the substrate density (the closest distance was measured to be ~ 14 Å away from the substrate), thus it is unclear from our model what their specific role may be. Notably, however, these positions (towards the N- and C-termini of the NTDs) are in flexible regions, thus we are not entirely confident of their precise location due to the resolution of the cryo-EM map and because they may be oriented differently in this active, hexamer state compared to the crystal structure. Moreover, given the conformational flexibility of the NTDs, different orientations may occur during the substrate translocation cycle that we are unable to capture but could potentially re-position these residues towards the substrate.

Nonetheless, **we thank the reviewer for pointing out these studies and have added the following statement and references to the Results section:**

“However, additional conserved residues (T7, D103, E109) that have previously been proposed to interact with the substrate (Liu, Tek et al. 2002, Barnett, Nagy

Model of NTD-trimer ring with triad residues (T7, D103, and E109) highlighted (cyan) and circled (dotted).

et al. 2005) appear more distal in our model, thus additional NTD conformations may be important during translocation.”

Reviewer #2

1. The authors test the importance of the proposed cross-loop interactions in NBD1, but they do not test the importance of some of the proposed interactions in NBD2. The authors propose that residues E639 and H641 interact directly with the substrate as it is exiting the channel and likely form additional stabilizing interactions that are important for substrate release. However, the authors do not mutate these residues and test disaggregation activity as they did for the cross-loop interactions. It would be ideal to see the impact of these mutations as well.

We have now mutated residues E639, K640 and H641 in the lower pore loop of NBD2 and tested function by the luciferase reactivation assay. All point mutants show a loss of activity relative to wildtype, supporting functional roles in substrate interaction and translocation that we propose from the structures. The loss of activity is significant (~40-60% of wt), but more modest compared to the primary pore loops, which appear to make stronger interactions with the substrate and thus are expected to be more essential to translocation. **These results have been added to Figure 3e and discussed in the Results section.**

2. The authors suggest that salt bridges formed by K250 (to E254) and R252 (to E256). The single mutations clearly show the importance of these basic residues, but complementing these charge inversions to restore the salt bridge would be a powerful validation of their model. Does the E254K or E356R restore activity in the K250E or R252E background?

We have now tested charge-reversal mutations at positions E254 (to K) and E256 (to R). Both point mutations show substantial loss of activity (~40% of WT for E254K and ~10% of Wt for E256R) supporting critical roles in disaggregation that we propose based on the structure. **These data have been added to Fig 2f and discussed in the Results.** We have also now performed substrate binding analysis by fluorescence polarization showing that charge reversal mutations at any of these 4 positions results in reduced binding to casein, further establishing their role in substrate interactions. **These data have been added to Supplementary Fig. 2b and discussed in the Results.** We have also tested the charge reversal double mutants at both positions (K250E:Q254K and R252E:E256R). We found that neither of these double mutants restored activity. We argue that these results do not rule out the presence of a salt bridge (or H-bond) between these residues because these residues are highly conserved among ClpB homologs and do not co-vary. **A diagram illustrating this conservation has been added to Supplementary Fig. 2c and discussed in the Results.** Thus, the surrounding environment, such as steric or charge constraints, may not allow for charge inversions to restore a functional salt bridge. Finally, in the recent ClpB structure from *M. tuberculosis* (published after our submission (Yu et al., Sept., 2018)) these same contacts are observed at a lower resolution, further substantiating our conclusions.

3. The authors suggest that the NTD ring plays a direct role in substrate binding and

transfer to the NBD domains based on their structure and on prior work (for example Rosezweig, et al) showing an important role for the NTD in substrate binding and activity. The surprising proposal here is the model that substrate recognition and processing uses only three NTDs (as a trimeric ring) as part of the disaggregation cycle, which the authors suggest is consistent with the enhanced activity seen in prior work with mixtures of ClpB isoforms. However, experimentally testing this model directly would be ideal. For example, do mutations on the surfaces of the NTDs that are not involved in the trimeric ring interface have no effect on substrate translocation? Can mutations that disrupt this trimeric structure (without effecting the A6/A1 helices) alter substrate translocation/recognition?

Mutagenesis of the NTD based on our structural model is an excellent suggestion and an important direction for further characterizing the function of NTD ring and the trimer architecture we identify. Considering that the mutagenesis approach would likely need to be exhaustive in order to identify mutations that specifically disrupt the NTD:NTD interface of the trimer but not other arrangements or substrate contacts, as the reviewer points out, we feel that these experiments are beyond the scope of our study here. Additionally, given that the presence of the NTD is not essential under certain *in vitro* disaggregation conditions, a number of *in vitro* and *in vivo* analyses would likely be required in order to identify distinct disaggregation steps or substrate interactions that are affected. Indeed, we are currently exploring a number of approaches for future studies to characterize the NTD trimer including mutagenesis, crosslinking-mass spec., and structural analysis of NTD-minus hetero-hexamer complexes.

4. Figures 1-2. The activity data is described as coming from two reactions (n=2), it would be better to simply show these two data points as a dot plot with the mean shown given the very small sample size.

We now show the two data points alongside the bar charts for these figures (1a, 2f, 3e).

Reviewer #3:

-Reviewer 3 questions the presence of side chain density for the substrate polypeptide in the cryo-EM map, indicating “there are no reports of Hsp100 substrate preference and the casein density will thus be an averaged density over all possible substrate processing steps.”

We agree that one would expect that the side-chain density for the casein substrate would be averaged out and not visible due to all the possible positions along the substrate ClpB could be bound under active translocation. However, we argue that in the cryo-EM map the density corresponding to the casein polypeptide in fact shows high-resolution features indicative of side-chain density, suggesting that the ClpB complexes may be bound to polypeptide sequences that are similar, thus enabling these higher-resolution features to be resolved in the map. To illustrate this point, we have included a comparison of a density map for a poly-gly strand, which would be a model for a featureless strand with averaged side chains, and the experimental cryo-EM map of the substrate density (see below). Densities are identified in the experimental map not present in the poly-gly strand that extend perpendicular from the main

backbone density in a regular $\sim 4\text{\AA}$ spacing consistent with amino acid side chains. Additionally, we found that although a model of a poly-Ala strand matched these side chain density positions, the fit was relatively poor when modeled with the NBD1 and NBD2 in Rosetta. This finding was, in fact, the rationale for fitting specific sequences of casein into the map - in order to achieve an improved fit and molecular model of the complex. While substrate sequence preferences for Hsp100s has not been extensively studied, Bukau and colleagues (Schlieker, Weibezahn et al. 2004) performed substrate binding analysis using a peptide library and identified that ClpB “discriminates between distinct amino acid side chains”, and binds to a specific subset of peptides that are enriched in aromatic residues and positively charged residues, while binding to peptides enriched in negatively charged residues are strongly disfavored. Our modeling analysis agrees with this study, indicating preferences for substrate sequences containing aromatic, hydrophobic, and to some extent, positively charged residues. **We have now included reference to this study in the Results.** Finally, it is worth mentioning that the use of the slowly hydrolysable analog, ATPyS may unleash or enhance substrate sequence binding specificities by ClpB given the high, nanomolar affinity for casein compared to ATP (see Gates et al., 2017). **We have included this statement in the Results.**

Figure showing simulated density for poly-glycine and extracted density from cryo-EM experimental map of the substrate polypeptide.

-Reviewer 3 indicates “It is surprising that the authors attempt to fit 1604 models of casein peptides into the density. It is very difficult to understand how the fits were weighted and thus how the energy scores were obtained”, and asks which scoring functions were used.

1604 sequences of casein were used because this provided full coverage of the 4 isoforms of bovine casein (α -s1, α -s2, β , and κ) that were present in the sample, and

included overlap between the peptides. **We have now added supplementary data (Supplementary Fig. 4) which plots the energies of all the peptides relative to poly-Alanine. We have expanded the explanation of the fits in the Methods section, which now states:**

The energy function used was Rosetta's most recent energy function, REF2015, and a weight of 65 on the density fitting term 'elec_dens_fast' was used. For each of the 1604 threaded models, 5 independent refinement trajectories were carried out, and the lowest-energy model was selected for analysis. From these sequence-energy pairs, a profile was constructed by computing – for each amino-acid at each position – the average energy over all sequences with the corresponding amino acid at the corresponding position. For this calculation, density scores were not used and only the Rosetta energy term was used to assess peptide energetics.

-Is there any indication that the modelled sequences are preferred by the enzyme in vitro?

As mentioned above, work by Bukau and colleagues (Schlieker, Weibezahn et al. 2004), identifies similar preferences for aromatic residues using an peptide-array analysis in vitro.

-What effect would the passage of a favorable binding sequence through ClpB have? Would the peptide bind stronger to the pore loops and passage would be slower?

Although we consider this too speculative to discuss in the manuscript, an ideal, low energy sequence based on our analysis might increase translocation processivity or impair substrate release.

-Reviewer 3 indicates that Suppl. Figure 5c (now Supplementary Fig 6d) is difficult to comprehend and requests that isolated densities should be shown with an explanation to support the proposed nucleotide state.

We thank the reviewer for pointing out the confusion of this figure. We have now enlarged and corrected the images of the nucleotide pockets of the seam protomers and show the density specifically for the nucleotide as a difference map (experimental map – apo model) as well as the positions of each Arginine finger side chain. Based on the density for nucleotide in these difference maps we propose the nucleotide state of each pocket. For the ATP states, the full ATP γ S molecule fits in the density, for the ADP states, the density is weaker and density likely corresponding to the γ -phosphate region is less resolved, and for the apo state, little to no density for nucleotide is observed. **These modifications have been made to Supplementary Fig. 6d and clarification has been made to the Results section (p. 12-13).**

-Reviewer 3 points out discrepancies in our proposed model for the function of the NBD1-NBD2 separation in the discussion and indicates that the “pulling mechanism” we describe does not fit with distance changes between domains given, in particular, that the increase in the separation only occurs when substrate is released and the protomers are in a post hydrolysis state.

We agree with the reviewer that our discussion of this proposed “pulling mechanism” needs clarification and thank the reviewer for the in-depth look at these conformational changes. **We have now revised this section of the Results (page 13-14) and included a model figure (Figure 7d) that connects the ATP hydrolysis cycle with NBD1-NBD2 conformational changes we observe. Additionally, we have further analyzed the conformational changes and also include a movie (Supplementary Movie 2) that shows a morph between these states.** We have clarified our discussion and do not argue that the NBD1-NBD2 changes contribute to a direct pulling mechanism because, as the reviewer points out, pulling could not occur once substrate release has happened. Because the conformational changes coincide with changes in the nucleotide state and substrate release at the seam, we now propose that they function in substrate binding and release steps and may help establish the clockwise rotary mechanism and translocation direction (N-terminal to C-terminal of ClpB)

-Reviewer 3 indicates “the manuscript lacks a clear comparison between the ClpBK467C (ClpBwt) and ClpBBAP variants” and asks for discussion as to why our structure of ClpBK476C lacks density for the middle domain.

The coiled-coil middle domain (MD) that wraps equatorially around the hexamer is flexible and thus difficult to resolve under the active translocation conditions we utilize. Indeed, we have previously identified for wt Hsp104 that the MD adopts two distinct nucleotide-specific conformations and that these two states co-exist in the casein bound complex. Notably, the coexistence of these states was a challenge to resolve, requiring in-depth 3D classification and they are at a lower resolution compared to the AAA+ core domains. The MD of ClpBBAP variants have only been identified to adopt what we argue is the ADP conformation of the MD (based on three structures of wt Hsp104 we have published, compared to Carroni et al, 2014 and Deville, et al., 2017 ClpBBAP analysis). With Reviewer 3’s important point about the MD, we have gone back and re-analyzed our 3D classification data and have identified the presence of MD density for the ClpBK467C complex bound to casein. **We are now able to resolve density for Motif 2 of the MD for protomers P3-P5 (the more stable, ATP-, substrate-bound protomers). We have included this data in Supplementary Fig 1I.** Based on the fit to the density we identify that the MD for these protomers matches the ATP conformation we previously characterized for Hsp104. This conformation is also identical to the MD conformation observed in the recent ClpB structure from *M. tuberculosis* (Yu et al., 2018) for the same, ATP-bound protomers. These data support our previously proposed model that the MD undergoes nucleotide specific conformational changes during active translocation and, for the ATP-bound protomers, Motif 2 contacts the adjacent, clockwise protomer across the nucleotide pocket of NBD1.

-Reviewer 3 minor points have been addressed:

- Line 114: the authors should clarify what they mean by the “stability was problematic under cryo conditions”. Was complex formation poor, did the protein get degraded or did the protein aggregate?

We are not sure of the source of the instability but found that the 2D averages were less well-resolved compared to ClpBK476C and the resolution of the 3D map was unable to improve beyond 5.7 Å. **We have clarified the sentence to read:**

“WT ClpB was initially tested and identified to form a stable substrate-bound complex by size exclusion chromatography (SEC) (Supplementary Fig. 1a), however initial reconstructions went to a modest, 5.7 Å-resolution, indicating hexamer instability or heterogeneity may be present (Supplementary Fig. 1b).”

- Line 116: an explanation of what “closed” state means is required rather than merely putting it in quotation marks.

We have added “substrate-bound” for clarification and referenced this statement.

- Line 333: this can only be the P6 Arg finger position

We thank the reviewer for pointing out the error and have changed the text indicate P6.

- Suppl. Figure 1h: The FSCs are truncated at ~4 Å. They need to be shown up to Nyquist frequency. The current diagram makes it impossible to assess whether the FSC resolution estimate is inflated. The FSC needs to drop to zero at high frequencies.

We thank the reviewer for pointing this out and have corrected the figure (**now Supplementary Fig. 1b**)

We have fixed all the spelling and text errors noted by Reviewer 3 and thank the reviewer for pointing these out.

Barnett, M. E., M. Nagy, S. Kedzierska and M. Zolkiewski (2005). "The amino-terminal domain of ClpB supports binding to strongly aggregated proteins." J Biol Chem **280**(41): 34940-34945.

Doyle, S. M., J. Shorter, M. Zolkiewski, J. R. Hoskins, S. Lindquist and S. Wickner (2007). "Asymmetric deceleration of ClpB or Hsp104 ATPase activity unleashes protein-remodeling activity." Nat Struct Mol Biol **14**(2): 114-122.

Hinnerwisch, J., W. A. Fenton, K. J. Furtak, G. W. Farr and A. L. Horwich (2005). "Loops in the central channel of ClpA chaperone mediate protein binding, unfolding, and translocation." Cell **121**(7): 1029-1041.

Johnston, D. M., M. Miot, J. R. Hoskins, S. Wickner and S. M. Doyle (2017). "Substrate Discrimination by ClpB and Hsp104." Front Mol Biosci **4**: 36.

Keiler, K. C., P. R. Waller and R. T. Sauer (1996). "Role of a peptide tagging system in degradation of proteins synthesized from damaged messenger RNA." Science **271**(5251): 990-993.

Li, T., C. L. Weaver, J. Lin, E. C. Duran, J. M. Miller and A. L. Lucius (2015). "Escherichia coli ClpB is a non-processive polypeptide translocase." Biochem J **470**(1): 39-52.

Liu, Z., V. Tek, V. Akoev and M. Zolkiewski (2002). "Conserved amino acid residues within the amino-terminal domain of ClpB are essential for the chaperone activity." J Mol Biol **321**(1): 111-120.

Nakazaki, Y. and Y. H. Watanabe (2014). "ClpB chaperone passively threads soluble denatured proteins through its central pore." Genes Cells **19**(12): 891-900.

Oguchi, Y., E. Kummer, F. Seyffer, M. Berynskyy, B. Anstett, R. Zahn, R. C. Wade, A. Mogk and B. Bukau (2012). "A tightly regulated molecular toggle controls AAA+ disaggregase." Nat Struct Mol Biol **19**(12): 1338-1346.

Schlieker, C., J. Weibezahn, H. Patzelt, P. Tessarz, C. Strub, K. Zeth, A. Erbse, J. Schneider-Mergener, J. W. Chin, P. G. Schultz, B. Bukau and A. Mogk (2004). "Substrate recognition by the AAA+ chaperone ClpB." Nat Struct Mol Biol **11**(7): 607-615.

Yu, H., T. J. Lupoli, A. Kovach, X. Meng, G. Zhao, C. F. Nathan and H. Li (2018). "ATP hydrolysis-coupled peptide translocation mechanism of Mycobacterium tuberculosis ClpB." Proc Natl Acad Sci U S A **115**(41): E9560-E9569.

REVIEWERS' COMMENTS:

Reviewer #1 (Remarks to the Author):

In the revised version, the Authors responded to all previous comments and included new experimental data. The manuscript is improved and can be recommended for publication after a minor revision:

1. Nowhere in the paper, I could find the source of ClpB used in this study. I am guessing that it is about ClpB from *Escherichia coli*, but that information should be included, given that some available structural data are for ClpB from *Thermus thermophilus*, *Mycobacterium tuberculosis*, and *Saccharomyces cerevisiae* (where it is called Hsp104, but it is an orthologous protein nevertheless).
2. Just a sentence about how ClpB was produced and purified is needed in Methods. This is important especially if any affinity tags were used for purification and not removed (N-terminal tags could interfere with substrate binding).
3. In the new Supplemental Fig. 1i, the horizontal axis is labeled "Hsp104 Hexamer". Was that Hsp104 from yeast and not the bacterial ClpB used in the rest of this study? If it was Hsp104, the experiment should be repeated for ClpB.

Reviewer #2 (Remarks to the Author):

The authors have addressed the majority of our concerns experimentally. While it would have been ideal to characterize the intriguing role of the NTD trimer more, it is understood that this is beyond the scope of this current work.

Samar Mahmoud
Peter Chien

Reviewer #3 (Remarks to the Author):

The manuscript is an improved version of the MS submitted originally. The authors have dealt with all referees' comments appropriately.

I have only one concern regarding comment 1 of reviewer 1. I disagree with his statement that the original ref 31 should not be used to support the location of the N-terminal domain. Although the structural model derived from the EM maps is controversial, the maps show clear density for the N-terminal domains. The location of the N-terminal domains is not a controversial aspect of this reference.

The sentence in lines 50-54 of the MS now reads: "In addition, amino-terminal domains (NTDs) are connected to the NBD1 by a flexible linker and form an additional ring in the hexamer that is important for interaction with some substrates (7,19,34-36), however the architecture of the NTD ring and specific functions during translocation are not well understood."

None of the cited references show that the NTDs form a ring. The hexameric complexes in reference 7 do not show any density for the N-terminal domains of ClpB, references 19 and 34 refer to substrate interaction, and reference 35 and 36 refer to the ClpB x-ray structure in ref7, which does not indicate ring formation of the N-terminal domains.

If the authors want to make the point that the N-terminal domains have been shown to interact with each other, they should cite a reference that proves this point. The original reference 31 was appropriate.

RESPONSE TO REVIEWERS' COMMENTS:

We thank the reviewers for their re-review of our manuscript and important comments. All comments have been addressed and are detailed below (blue).

Reviewer #1 (Remarks to the Author):

In the revised version, the Authors responded to all previous comments and included new experimental data. The manuscript is improved and can be recommended for publication after a minor revision:

1. Nowhere in the paper, I could find the source of ClpB used in this study. I am guessing that it is about ClpB from *Echerichia coli*, but that information should be included, given that some available structural data are for ClpB from *Thermus thermophilus*, *Mycobacterium tuberculosis*, and *Saccharomyces cerevisiae* (where it is called Hsp104, but it is an orthologous protein nevertheless).

We thank the reviewer for pointing this out and have added reference to the use of ClpB from *Escherichia coli* to the Introduction and to the Methods sections.

2. Just a sentence about how ClpB was produced and purified is needed in Methods. This is important especially if any affinity tags were used for purification and not removed (N-terminal tags could interfere with substrate binding).

We have included a brief description of the purification procedure in the Methods. Notably, a C-terminal His6-tag was used for affinity purification.

3. In the new Supplemental Fig. 1i, the horizontal axis is labeled "Hsp104 Hexamer". Was that Hsp104 from yeast and not the bacterial ClpB used in the rest of this study? If it was Hsp104, the experiment should be repeated for ClpB.

We thank the reviewer for pointing out this error. This was a typo, and has been changed to ClpB, which is the protein that was used for the binding analysis

Reviewer #2 (Remarks to the Author):

The authors have addressed the majority of our concerns experimentally. While it would have been ideal to characterize the intriguing role of the NTD trimer more, it is understood that this is beyond the scope of this current work.

Samar Mahmoud
Peter Chien

Reviewer #3 (Remarks to the Author):

The manuscript is an improved version of the MS submitted originally. The authors have dealt with all referees' comments appropriately.

I have only one concern regarding comment 1 of reviewer 1. I disagree with his statement that the original ref 31 should not be used to support the location of the N-terminal domain. Although the structural model derived from the EM maps is controversial, the maps show clear density for

the N-terminal domains. The location of the N-terminal domains is not a controversial aspect of this reference.

The sentence in lines 50-54 of the MS now reads: "In addition, amino-terminal domains (NTDs) are connected to the NBD1 by a flexible linker and form an additional ring in the hexamer that is important for interaction with some substrates (7,19,34-36), however the architecture of the NTD ring and specific functions during translocation are not well understood."

None of the cited references show that the NTDs form a ring. The hexameric complexes in reference 7 do not show any density for the N-terminal domains of ClpB, references 19 and 34 refer to substrate interaction, and reference 35 and 36 refer to the ClpB x-ray structure in ref7, which does not indicate ring formation of the N-terminal domains.

If the authors want to make the point that the N-terminal domains have been shown to interact with each other, they should cite a reference that proves this point. The original reference 31 was appropriate.

We agree with the reviewer and have added the Wendler et al., citation to reference the NTD architecture